# LEARNING WITH NOISY LABELS REVISITED: A STUDY USING REAL-WORLD HUMAN ANNOTATIONS

**Jiaheng Wei**[*][†]**, Zhaowei Zhu**[*][†]**, Hao Cheng**[†]**, Tongliang Liu**[‡]**, Gang Niu**[§]**, and Yang Liu**[†]

[†]University of California, Santa Cruz,   [‡]TML Lab, University of Sydney,   [§]RIKEN

[†]`{jiahengwei,zwzhu,haocheng,yangliu}@ucsc.edu,`
[‡]`tongliang.liu@sydney.edu.au,`   [§]`gang.niu.ml@gmail.com`

## ABSTRACT

Existing research on learning with noisy labels mainly focuses on synthetic label noise. The synthetic noise, though has clean structures which greatly enabled statistical analyses, often fails to model the real-world noise patterns. The recent literature has observed several efforts to offer real-world noisy datasets, e.g., Food-101N, WebVision, and Clothing1M. Yet the existing efforts suffer from two caveats: firstly, the lack of ground-truth verification makes it hard to theoretically study the property and treatment of real-world label noise. Secondly, these efforts are often of large scales, which may result in unfair comparisons of robust methods within reasonable and accessible computation power. To better understand real-world label noise, it is important to establish controllable, easy-to-use and moderate-sized real-world noisy datasets with both ground-truth and noisy labels. This work presents two new benchmark datasets, which we name as CIFAR-10N, CIFAR-100N (jointly we call them CIFAR-N), equipping the training datasets of CIFAR-10 and CIFAR-100 with human-annotated real-world noisy labels we collected from Amazon Mechanical Turk. We quantitatively and qualitatively show that real-world noisy labels follow an instance-dependent pattern rather than the classically assumed and adopted ones (e.g., class-dependent label noise). We then initiate an effort to benchmarking a subset of the existing solutions using CIFAR-10N and CIFAR-100N. We further proceed to study the memorization of correct and wrong predictions, which further illustrates the difference between human noise and class-dependent synthetic noise. We show indeed the real-world noise patterns impose new and outstanding challenges as compared to synthetic label noise. These observations require us to rethink the treatment of noisy labels, and we hope the availability of these two datasets would facilitate the development and evaluation of future learning with noisy label solutions. The corresponding datasets and the leaderboard are available at `http://noisylabels.com`.

## 1 INTRODUCTION

Image classification task in deep learning requires assigning labels to specific images. Annotating labels for training use often requires tremendous expenses on the payment for hiring human annotators. The pervasive noisy labels from data annotation present significant challenges to training a quality machine learning model. The problem of dealing with label noise has been receiving increasing attentions. Typical approaches include unbiased estimators and weighted loss functions (Natarajan et al., 2013; Liu & Tao, 2015), loss correction (Patrini et al., 2017; Liu & Guo, 2020), sample selection aided (Jiang et al., 2018; Han et al., 2018; Yu et al., 2019), etc. The majority of existing solutions are often developed under stylish synthetic noise model, where the noise rates are either class-dependent or homogeneous across data instances. However, real-world supervision biases may come from humans (Peterson et al., 2019), sensors (Wang et al., 2021b), or models (Zhu et al., 2022), which are likely to be instance-dependent. Recent works on instance-dependent settings (Cheng et al., 2021; Jiang et al., 2022) also have some structural assumptions, e.g., the noise transition differs

---

[*]Equal contributions in alphabetical ordering.

[†]Corresponding author: Yang Liu <yangliu@ucsc.edu>.

Table 1: Summarized information of existed noisy-label benchmarks: the "estimated" noisy levels are obtained through a subset of the dataset with verified clean labels. "Moderate-resolution" means the max image width pixel is less than 250.

| Dataset | Train/Test Size | Classes | Noise level | Moderate resolution | Clean label | No Interventions |
|---|---|---|---|---|---|---|
| **Food-101** (Bossard et al., 2014) | 75.75K / 25.25K | 101 | N/A | ✗ | ✗ | ✓ |
| **Clothing1M** (Xiao et al., 2015) | 1M in all | 14 | Estimated 38% | ✗ | ✗ | ✓ |
| **WebVision** (Li et al., 2017) | ≈2.44M / 100K | 1000 | N/A | ✗ | ✗ | ✓ |
| **Food-101N** (Lee et al., 2018) | 310K in all | 101 | Estimated 20% | ✗ | ✗ | ✓ |
| **Animal-10N** (Song et al., 2019) | 50K / 5K | 10 | 8% | ✓ | ✓ | ✗ |
| **Red Mini-ImageNet** (Jiang et al., 2020) | 50K / 5K | 100 | 0%-80% | ✓ | ✓ | ✗ |
| **Red Stanford Cars** (Jiang et al., 2020) | 8K / 8K | 196 | 0%-80% | ✓ | ✓ | ✗ |
| **CIFAR10H** (Peterson et al., 2019) | 50K / 10K | 10 | N/A | ✓ | ✓ | ✓ |
| **CIFAR-10N-aggregate (Ours)** | 50K / 10K | 10 | 9.03% | ✓ | ✓ | ✓ |
| **CIFAR-10N-random (Ours)** | 50K / 10K | 10 | ≈18% | ✓ | ✓ | ✓ |
| **CIFAR-10N-worse (Ours)** | 50K / 10K | 10 | 40.21% | ✓ | ✓ | ✓ |
| **CIFAR-100N-coarse (Ours)** | 50K / 10K | 20 | 25.60% | ✓ | ✓ | ✓ |
| **CIFAR-100N-fine (Ours)** | 50K / 10K | 100 | 40.20% | ✓ | ✓ | ✓ |

in different parts of features (Xia et al., 2020b) or sub-populations (Wang et al., 2021a; Zhu et al., 2021a). Although these statistical assumptions facilitate the derivation of theoretical solutions, it is unclear how the existing models captured the real-world noise scenario.

To empirically validate the robustness of proposed methods, synthetic noisy labels on CIFAR-10 and CIFAR-100 (Krizhevsky et al., 2009) are the most widely accepted benchmarks. The literature has also observed approaches to the simulation of human annotators in data labeling (Hua et al., 2013; Long & Hua, 2015; Liao et al., 2021), and real-world label noise benchmarks, including Food-101 (Bossard et al., 2014), Clothing-1M (Xiao et al., 2015), WebVision (Li et al., 2017), etc. We summarize the above real-world noisy label datasets in Table 1. While a more detailed description and discussion of the existing datasets can be found in the related works, we want to highlight several outstanding issues in existing benchmarks and evaluations. As noted in Table 1, except for CIFAR related noisy label datasets, all other datasets suffer from at least one of the three caveats:

- Complex task (High-resolution): when learning with large-scale and relative high-resolution data, the complex data pattern, various augmentation strategies (Xiao et al., 2015), the use of extra train or clean data (Bossard et al., 2014; Xiao et al., 2015; Lee et al., 2018), different computation power (for hyper-parameter tuning such as batch-size, learning rate, etc) jointly contribute to the model performance and then result in unfair comparison.

- Missing clean labels: the lack of clean labels for verification in most existed noisy-label datasets makes the evaluation of robust methods intractable.

- Interventions: human interventions in data generation (Jiang et al., 2020) and non-representative data collection process (Song et al., 2019) might disturb the original noisy label pattern.

In addition, despite synthetically labeled CIFAR datasets are popular and highly used benchmarks for evaluating the robustness of proposed methods, there exists no publicly available human annotated labels for CIFAR training datasets to perform either validation of existing methods or verification of popular noise models [1]. A human-annotated version of CIFAR datasets would greatly facilitate the evaluations of existing and future solutions, due to the already standardized procedures for experimenting with CIFAR datasets. All above issues motivate us to revisit the problem of learning with noisy labels and establish accessible and easy-to-use, verifiable datasets that would be broadly usable to the research community. Our contributions can be summarized as follows :

- We present two new benchmarks CIFAR-10N, CIFAR-100N which provide CIFAR-10 and CIFAR-100 with human annotated noisy labels. Jointly we call our datasets CIFAR-N. Our efforts built upon the CIFAR datasets and provide easily usable benchmark data for the weakly supervised learning community (Section 3). We expect to continue to maintain the datasets to facilitate future development of results.

- We introduce new observations for the distribution of human annotated noisy labels on tiny images, i.e., imbalanced annotations, the flipping of noisy labels among similar features, co-existence of multiple clean labels for CIFAR-100 train images (which leads to a new pattern of label noise), etc. We further distinguish noisy labels in CIFAR-10N and CIFAR-100N with synthetic class-dependent label noise, from the aspect of noise transitions for different features qualitatively and quantitatively (via hypothesis testing) (Section 4).

---

[1]CIFAR10H (Peterson et al., 2019) only provides test images with noisy human annotations.

- We empirically compare the robustness of a comprehensive list of popular methods when learning with CIFAR-10N, CIFAR-100N. We observe consistent performance gaps between human noise and synthetic noise. The different memorization behavior further distinguishes the human noise and synthetic noise (Section 5). The corresponding datasets and the leaderboard are publicly available at `http://noisylabels.com`.

### 1.1 RELATED WORKS

**Learning from noisy labels**  Earlier approaches for learning from noisy labels mainly focus on loss adjustment techniques. To mitigate the impact of label noise, a line of approaches modify the loss of image samples by multiplying an estimated noise transition matrix (Patrini et al., 2017; Hendrycks et al., 2018; Xia et al., 2019; Yao et al., 2020), re-weight the loss to encourage deep neural nets to fit on correct labels (Liu & Tao, 2015), propose robust loss functions (Natarajan et al., 2013; Ghosh et al., 2017; Zhang & Sabuncu, 2018; Amid et al., 2019; Wang et al., 2019; Liu & Guo, 2020), or introduce a robust regularizer (Liu et al., 2020; Xia et al., 2020a; Cheng et al., 2021; Wei et al., 2021). Another line of popular approaches behaves like a semi-supervised manner which begins with a clean sample selection procedure, then makes use of the wrongly-labeled samples. For example, several methods (Jiang et al., 2018; Han et al., 2018; Yu et al., 2019; Wei et al., 2020) adopt a mentor/peer network to select small-loss samples as "clean" ones for the student/peer network. To further explore the benefits of wrongly-labeled samples and improve the model performance, Li et al. (2020) chose the MixMatch (Berthelot et al., 2019) technique which has shown success in semi-supervised learning.

**Benchmarks noisy labels datasets**  Food-101 (Bossard et al., 2014), Clothing-1M (Xiao et al., 2015), WebVision (Li et al., 2017) are three large-scale noisy labeled web-image databases which consist of food images, clothes images or other web images, respectively. However, the majority of images in these three datasets do not have a corresponding clean label to perform controlled verification (e.g., verifying the noise levels). Later, a much larger-scale food dataset is collected by Lee et al. (2018), which contains exactly the same classes as Food-101 (Bossard et al., 2014). More recently, Peterson et al. (2019) present a noisily labeled benchmarks on CIFAR-10 test dataset where each test image has 51 human annotated labels in average. Jiang et al. (2020) construct noisily labeled Mini-ImageNet (Vinyals et al., 2016) and Stanford Cars datasets (Krause et al., 2013) with controlled noise levels by substituting human annotated incorrect labels for synthetic wrong labels.

## 2 SYNTHETIC LABEL NOISE

In this section, we discuss a few popular synthetic models for generating noisy labels. We focus on a $K$-class classification task. Denote by $D := \{(x_n, y_n)\}_{n \in [N]}$ the training samples where $[N] := \{1, 2, ..., N\}$. $(x_n, y_n)$s are given by random variables $(X, Y) \in \mathcal{X} \times \mathcal{Y}$ drawn from the joint distribution $\mathcal{D}$, where $\mathcal{X}, \mathcal{Y}$ can be viewed as the space of feature and label, respectively. In real-world scenarios, a classifier $f$ only has access to noisily labeled training sets $\widetilde{D} := \{(x_n, \tilde{y}_n)\}_{n \in [N]}$. We assume the noisy samples $(x_n, \tilde{y}_n)$s are given by random variables $(X, \widetilde{Y}) \in \mathcal{X} \times \widetilde{\mathcal{Y}}$ which are drawn from the joint distribution $\widetilde{\mathcal{D}}$. Clearly, there may exist $n \in [N]$ such that $y_n \neq \tilde{y}_n$. The flipping from clean to noisy label is usually formulated by a noise transition matrix $T(X)$, with elements: $T_{i,j}(X) := \mathbb{P}(\widetilde{Y} = j|Y = i, X)$. We shall specify different modeling choices of $T(X)$ below.

### 2.1 CLASS-DEPENDENT LABEL NOISE

The first family of noise transition matrix is the class-dependent noise where the label noise is assumed to be conditionally independent of the feature $X$. Mathematically, $T(X) \equiv T$ and

$$T_{i,j}(X) = \mathbb{P}(\widetilde{Y} = j|Y = i), \forall i, j \in [K].$$

**Symmetric $T$**  The symmetric noise transition matrix (Natarajan et al., 2013) describes the scenario where an amount of human labelers maliciously assign a random label for the given task. It assumes that the probability of randomly flipping the clean class to the other possible class with probability $\epsilon$. Assume the noise level is $\epsilon$, the diagonal entry of the symmetric $T$ is denoted as $T_{i,i} = 1 - \epsilon$. For any other off-diagonal entry $T_{i,j}$ where $i \neq j$, the corresponding element is $T_{i,j} = \frac{\epsilon}{K-1}$.

**Asymmetric** $T$    The asymmetric noise transition matrix (Patrini et al., 2017) simulates the case where there exists ambiguity classes, i.e, human labelers may wrongly annotate the truck as automobile due to the low-resolution images. There are two types of widely adopted asymmetric $T$. The assymetric-next $T$ assumes that the clean label flips to the next class with probability $\epsilon$, i.e, $i \to (i+1) \mod K$ for $i \in [K]$. The assymetric-pair $T$ considers $[\frac{K}{2}]$ disjoint class pairs $(i_c, j_c)$ where $i_c < j_c$. For $c \in [\frac{K}{2}]$, $T_{i_c,j_c} = T_{j_c,i_c} = \epsilon$, and the diagonal entries are $1 - \epsilon$.

## 2.2    Instance-dependent label noise

Beyond the feature-independent assumption, recent works pay more attention to a challenging case where the label noise is jointly determined by feature $X$ and clean label $Y$. There are some techniques for synthesizing the instance-dependent label noise, such as the polynomial margin diminishing label noise (Zhang et al., 2021b) where instances near decision boundary are easier to be mislabeled, the part-dependent label noise (Xia et al., 2020b) where different parts of feature may contribute different noise transition matrices, and the group-dependent label noise (Wang et al., 2021a; Zhu et al., 2021a) where different sub-populations may have different noise rates. All of these noise models are proposed with some statistical assumptions, which facilitate the derivation of theoretical solutions.

## 3    Human annotated noisy labels on CIFAR-10, CIFAR-100

In this section, we introduce two new benchmark datasets for learning with noisy labels: CIFAR-10N and CIFAR-100N. Both datasets are built using human annotated labels collected on Amazon Mechanical Turk (M-Turk): we post CIFAR-10 and CIFAR-100 training images as the annotation Human Intelligence Tasks (HITs), and workers receive payments by completing HITs.

### 3.1    CIFAR-10N real-world noisy label benchmark

CIFAR-10 (Krizhevsky et al., 2009) dataset contains 60k $32 \times 32$ color images, 50k images for training and 10k images for testing. Each image belongs to one of ten completely mutually exclusive classes: airplane, automobile, bird, cat, deer, dog, frog, horse, ship, and truck.

**Dataset collection**    We randomly split the training dataset of CIFAR-10 without replacement into ten batches. In the Mturk interface, each batch contains 500 HITs with 10 images per HIT. The training images and test dataset remain unchanged. Each HIT is then randomly assigned to three independent workers. Workers gain base reward $0.03 after submitting the answers of each HIT. We reward workers with huge bonus salary if the worker contributes more HITs than the averaged number of submissions. We did not make use of any ground-truth clean labels to approve or reject submissions. We only block and reject workers who submit answers with fixed/regular distribution patterns. We defer more details of the dataset collection to Appendix A.

**Dataset statistics**    For CIFAR-10N dataset, each training image contains one clean label and three human annotated labels. We provide five noisy-label sets as follows.

- **Aggregate:** aggregation of three noisy labels by majority voting. If the submitted three labels are different for an image, the aggregated label will be randomly selected among the three labels.
- **Random** $i$ ($i \in \{1, 2, 3\}$)**:** the $i$-th submitted label for each image. Note our collection procedure ensures that one image cannot be repeatedly labeled by the same worker.
- **Worst:** dataset with the highest noise rate. For each image, if there exist any wrongly annotated labels in three noisy labels, the worst label is randomly selected from wrong labels. Otherwise, the worst label is equal to the clean label.

In CIFAR-10N, **60.27%** of the training images have received unanimous label from three independent labelers. The noise rates of prepared five noisy label sets are **9.03% (Aggregate), 17.23% (Random 1), 18.12% (Random 2), 17.64% (Random 3)** and **40.21% (Worst)**. A complete dataset comparison among existing benchmarks and ours are given in Table 1. We defer the noise level of each batch to Table 4 (Appendix). Aggregating the annotated labels significantly decreases the noise rates. All three random sets have $\approx 18\%$ noise level. To provide a challenging noisy setting, we also prepare a worst label set which serves to cover highly possible mistakes from human annotators on CIFAR-10.

## 3.2 CIFAR-100N REAL-WORLD NOISY LABEL BENCHMARK

CIFAR-100 (Krizhevsky et al., 2009) dataset contains 60K $32 \times 32$ color images of 100 fine classes, 50000 images for training and 10000 images for testing. Each (fine) class contains 500 training images and 100 test images. The 100 classes are grouped into 20 mutually exclusive super-classes.

**Data collection** We split the training dataset of CIFAR-100 without replacement into ten batches with five images per HIT. Only one worker is assigned to each HIT. We group the 100 classes into 20 disjoint super-classes (see Table 5 in the Appendix) which are slightly different from the 20 "coarse" categories named in CIFAR-100. The fine label in each super-class is summarized in Table 6 (Appendix). Workers are instructed to firstly select the super-class for each image. And will then be re-directed to the best matched fine label. Since some super-classes are hard to recognize without prior knowledge in biology, we provide workers with easy access to re-select the super-class, and every fine class has an example image for references. The rejecting rule and the bonus policy are the same as those in CIFAR-10N. We defer more details of the dataset collection to Appendix B.

**Dataset statistics** For CIFAR-100N dataset, each image contains a coarse label and a fine label given by a human annotator. Most batches have approximately **40%** noisy fine labels and **25 %** noisy coarse labels. The overall noise level of coarse and fine labels are **25.60%** and **40.20%**, respectively. We defer the noise level of each batch to Table 7 (Appendix).

## 4 PRELIMINARY OBSERVATIONS ON CIFAR-10N, CIFAR-100N

In this section, we analyze the human annotated labels for CIFAR-10 and CIFAR-100. We will empirically compare the noisy labels in CIFAR-10N and CIFAR-100N with class-dependent label noise both qualitatively and quantitatively.

### 4.1 THE NOISY LABEL DISTRIBUTION

**Observation 1: Imbalanced annotations** Our first observation is the imbalanced contribution of labels. Note that while the number of images are the same for each clean label, across all the five noisy label sets of CIFAR-10N, we observe that human annotators have different preferences for similar classes. For instance, they are more likely to annotate an image to be an automobile rather than the truck, to be the horse rather than the deer (see Figure 8 in the Appendix). The aggregated labels appear more frequently in automobile and ship, and less frequently in deer and cat. This gap of frequency becomes more clear in the worst label set. In CIFAR-100N, human annotators annotate frequently on classes which are outside of the clean-coarse, i.e., 25% noisy labels fall outside of the super-class and 15% inside the super-class. And the phenomenon of imbalanced annotations also appears substantially as shown in Figure 1, which presents the distribution of noisy labels for each selected fine class. "Man" appears $\geq 750$ times, while "Streetcar" only has $\approx 200$ annotations.

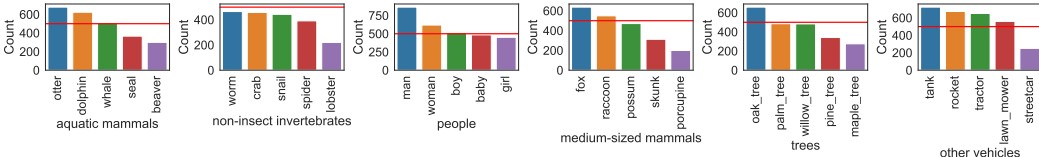

Figure 1: Categorical distribution of noisy labels on CIFAR-100N (most imbalanced 6 super-classes): the red line in each subplot indicates that the number of each clean fine class is 500.

**Observation 2: Noisy label flips to similar features** In CIFAR-100N, most fine classes are more likely to be mislabeled into less than four fine classes. In Figure 2, we show top three wrongly annotated fine labels for several fine classes that have a relative large noise rate. Due to the low-resolution of images, a number of noisy labels are annotated in pair of classes, i.e, $\approx 20\%$ of "snake" and "worm" images are mislabeled between each other, similarly for "cockroack"-"beetle", "fox"-"wolve", etc. While some other noisy labels are more frequently annotated within more classes, such as "boy"-"baby"-"girl"-"man", "shark"-"whale"-"dolphin"-"trout", etc, which share similar features.

**Observation 3: The pattern of noise transition matrices** In the class-dependent label noise setting, suppose the label noise is conditional independent of the feature, the noise transition matrices of CIFAR-10N and CIFAR-100N are best described by a mixture of symmetric and asymmetric $T$. For CIFAR-10N, we heatmap the aggregated noisy labels, random1 noisy labels and worst noise

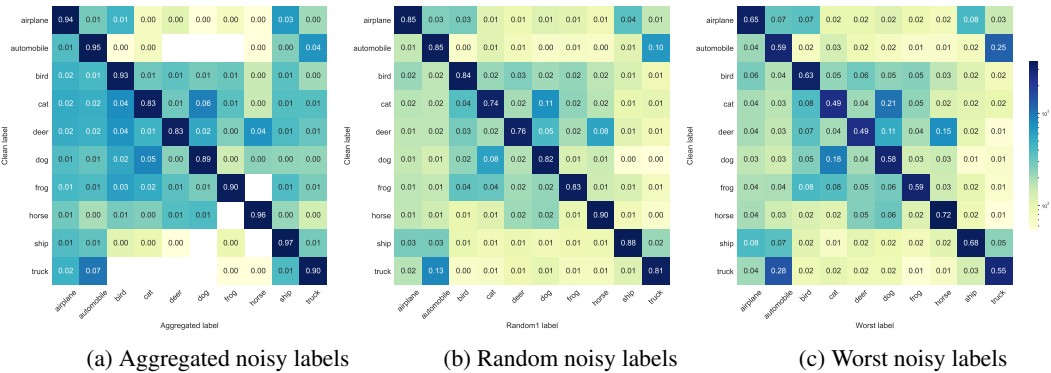

Figure 2: Top 3 wrongly annotated fine labels in selected fine classes. For "pine tree", "shrew", "streetcar", the dominant class is the **wrong** class. The corresponding number of correct annotations are highlighted with red lines.

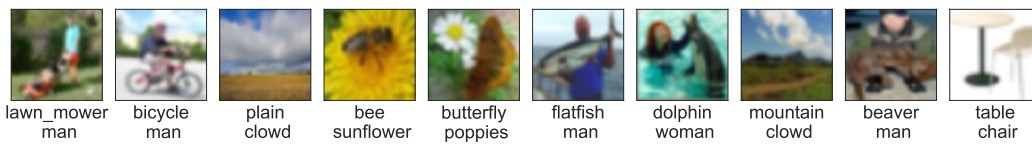

(a) Aggregated noisy labels     (b) Random noisy labels     (c) Worst noisy labels

Figure 3: Transition matrix of CIFAR-10N noisy labels (color bar is log-norm transformed).

labels w.r.t. the clean labels. In Figure 3, the three noisy label sets share a common pattern: the clean label flips into one or more similar classes more often. The remaining classes largely follow the symmetric noise model with a low noise rate. For example, "truck" and "automobile" flip between each other more often ($\approx 25\% - 30\%$ percentage), which is much larger than that of all other classes. Besides, in the central area of each transition matrix, it is quite obvious that the clean label of animal classes flips more often to other animals. Similar observations hold in CIFAR-100N, where each class flips to a few misleading classes with much higher probability than that of remaining ones (see Figure 11 in the Appendix). Apparently, current synthetic class-dependent noisy settings are not as complex as the real-world human annotated label noise.

**Observation 4: Label noise: bad news or good news?** During the label collection, there exist a non-negligible amount of the wrongly annotated classes that indeed co-exist in the corresponding images. In other words, training images of CIFAR-100 may contain **multiple labels** rather than a single one. We select several exemplary training images of CIFAR-100 where multiple labels appear (in Figure 4). The annotated class also appears in the corresponding image while is deemed as a wrong annotation by referring to the officially provided clean label. The most frequent case is best described by the scenario where a man holding a flatfish in hands. The clean label usually comes to "flatfish", while human annotators are more likely to categorize these images into "man". We conjecture that with the increasing label dimension, the phenomenon of multiple clean labels might be a more common issue. We leave more explorations for the future work.

Figure 4: Exemplary CIFAR-100 training images with multiple labels. The text below each picture denotes the CIFAR-100 clean label (first row) and the human annotated noisy label (second row).

## 4.2 HUMAN NOISY LABELS V.S. SYNTHETIC NOISY LABELS

Recall that, for the class-dependent label noise, we have $\mathbb{P}(\widetilde{Y}|X, Y) = \mathbb{P}(\widetilde{Y}|Y)$, indicating the noise transitions are identical for different features. For general instance-dependent label noise, the above equality may not hold. In this subsection, we explore to what degree this feature-dependency holds by checking whether the equality is satisfied or not for different features. In the following, by checking

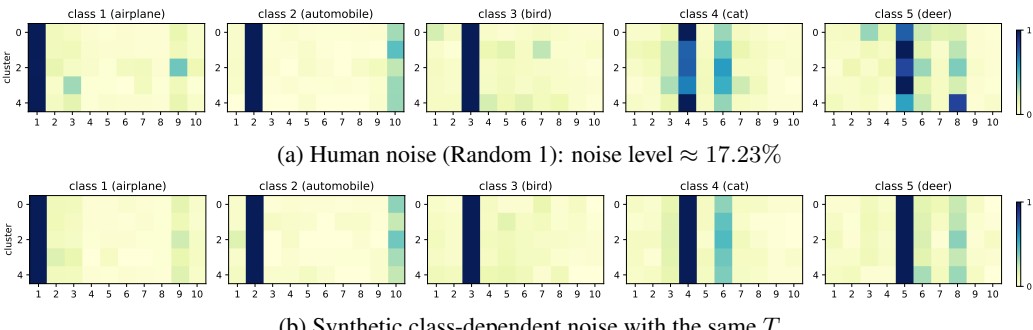

(a) Human noise (Random 1): noise level $\approx 17.23\%$

(b) Synthetic class-dependent noise with the same $T$

Figure 5: Illustration of noise transitions of human-level label noise and the synthetic version. We divide the representations of images from the same true class into 5 clusters by $k$-means. The representations come from the output before the final fully-connected layer of ResNet34. The model is trained on clean CIFAR-10. Negative cosine similarity measures the distance between features.

the equality for different features, **we will show the feature-dependency from both a qualitative aspect and a quantitative aspect.**

### 4.2.1 A QUALITATIVE ASPECT

We first visualize the noise transitions for different features from a qualitative aspect. Taking CIFAR-10N as an example, each image in CIFAR-10 only appears once. Without additional assumptions, we only have three noisy labels for each individual feature $X$, which makes it difficult to accurately estimate the noise transition probability $T(X), \forall X$, even though the ground-truth clean label is available. To make the estimation tractable, we consider estimations on locally homogeneous label noise, i.e., nearby features share the same $T(X)$. See the rigorous definition as follows.

**Definition 1** ($M$-**NN noise clusterability**) *(Zhu et al., 2021b) We call $\widetilde{\mathcal{D}}_n$ satisfies $M$-NN ($M$-Nearest-Neighbor) noise clusterability if the $M$-NN of $x_n$ have the same noise transition matrix as $x_n$, i.e., $T(x_n) = T(x_{n_i}), \forall i \in [M]$ where $\{x_{n_i}\}_{i \in [M]}$ denote the $M$-NN of $x_n$.*

With $M$-NN noise clusterability, we can estimate $T(X)$ on each subset $\widetilde{\mathcal{D}}_n$. In our visualization (e.g., Figure 5), rather than manually fix a particular $M$, we use the $k$-means algorithm to separate features belonging to the same true class to 5 clusters and adaptively find a suitable $M$ for each cluster. Denote by $\mathcal{I}_{i,\nu}$ the set of instance indices from the $\nu$-th cluster of clean class $i$. Then for label noise in $\mathcal{I}_{i,\nu}$, we assume it is feature-independent and denote the corresponding transition vector by $\boldsymbol{p}_{i,\nu}$, where each element $\boldsymbol{p}_{i,\nu}[j]$ is expected to be $\mathbb{P}(\widetilde{Y} = j | x_n, n \in \mathcal{I}_{i,\nu}, Y = i)$. The transition vector $\boldsymbol{p}_{i,\nu}$ can be estimated by counting the frequency of each noisy class given noisy labels in $\mathcal{I}_{i,\nu}$. For example, in Figure 5a, each row of the top-right subfigure titled "Class 5 (deer)" shows $\mathbb{P}(\widetilde{Y}|X, Y = 5)$ for different clusters of $X$, i.e., transition vectors $\boldsymbol{p}_{5,\nu}, \nu \in \{1, \cdots, 5\}$. Before putting up a more formal testing, we clearly observe that different transition vectors across different feature clusters, signaling the fact that the noise transitions $\mathbb{P}(\widetilde{Y}|X, Y = 5)$ are feature-dependent. For a controlled comparison, we synthesize the class-dependent label noise with the same expected noised transition matrix $T := \mathbb{E}[T(X)]$ as our human-level label noise and illustrate it in Figure 5b. We can find that different rows of each matrix in Figure 5b are very similar, showing the synthetic noise is feature-independent and our verification method is valid.

### 4.2.2 A QUANTITATIVE ASPECT

In this section, we quantitatively compare human noise and synthetic class-dependent noise through hypothesis testing.

**Formulation** Following the clustering results in Section 4.2.1, we further statistically test whether the human-noise is feature-dependent or not. For CIFAR-10N, the null hypothesis $\mathbf{H_0}$ and the corresponding alternate hypothesis $\mathbf{H_1}$ are defined as:

$\mathbf{H_0}$ : Human annotated label noise in CIFAR-10N is *feature-independent*;

$\mathbf{H_1}$ : Human annotated label noise in CIFAR-10N is *feature-dependent*.

Note that the synthetic label noise illustrated in Figure 5b is supposed to be feature-independent, the above hypotheses are converted to:

$\mathbf{H_0}$ : Human annotated label noise in CIFAR-10N is *the same as* the corresponding synthetic one;

$\mathbf{H_1}$ : Human annotated label noise in CIFAR-10N is *different from* the corresponding synthetic one.

From Figure 5, one measure of the difference between human noise and synthetic noise is the distance between transition vectors $\boldsymbol{p}_{i,\nu}$ across different noise, e.g., $d_{i,\nu}^{(1)} := \|\boldsymbol{p}_{i,\nu}^{\text{human}} - \boldsymbol{p}_{i,\nu}^{\text{synthetic}}\|_2^2$. As contrast, we need to compare $d_{i,\nu}^{(1)}$ with $d_{i,\nu}^{(2)} := \|\boldsymbol{p}_{i,\nu}^{\text{synthetic}'} - \boldsymbol{p}_{i,\nu}^{\text{synthetic}}\|_2^2$, where $\boldsymbol{p}_{i,\nu}^{\text{synthetic}'}$ denotes the transition vector from the same synthetic noise but different clustering result (caused by random data augmentation). Intuitively, if $d_{i,\nu}^{(1)}$ is much greater than $d_{i,\nu}^{(2)}$, we should accept $\mathbf{H_1}$. The above hypotheses are then equivalent to:

$\mathbf{H_0}$ : $\{d_{i,\nu}^{(1)}\}_{i\in[K],\nu\in[5]}$ come from the *same* distribution as $\{d_{i,\nu}^{(2)}\}_{i\in[K],\nu\in[5]}$;

$\mathbf{H_1}$ : $\{d_{i,\nu}^{(1)}\}_{i\in[K],\nu\in[5]}$ come from *different* distributions from $\{d_{i,\nu}^{(2)}\}_{i\in[K],\nu\in[5]}$.

We repeat the generation of $d_{i,\nu}$ for 10 times, where the images are modified with different data augmentations each time. We choose the significance level $\alpha = 0.05$ and perform a two-sided t-test w.r.t $\{d_{i,\nu}^{(1)}\}_{i\in[K],\nu\in[5]}$ and $\{d_{i,\nu}^{(2)}\}_{i\in[K],\nu\in[5]}$. Hypothesis testing results show that $p = 1.8e^{-36}$ for the whole data. Thus, the null hypothesis is rejected with the significance value $\alpha$, hypothesis "**human annotated label noise in CIFAR-10N is *feature-dependent***" is accepted. Similar conclusion can be reached for CIFAR-100N, we defer more details to Appendix D.

## 5 LEARNING WITH CIFAR-10N AND CIFAR-100N

We reproduce several popular and state-of-the-art robust methods on synthetic noisy labels (using calculated transition matrices from our collected data) as well as our collected human annotated labels. Most of the selected methods fall into loss correction, loss re-weighting, and loss regularization related methods, etc.

### 5.1 PERFORMANCE COMPARISONS ON CIFAR-10N AND CIFAR-100N

For a fair comparison, we adopt ResNet-34 (He et al., 2016), the same training procedure and batch-size for all implemented methods. More experiment details are deferred to the Appendix E. In Table 2, note that both ELR+ (Liu et al., 2020) and Divide-Mix (Li et al., 2020) adopt two networks with advanced strategies such as mix-up data augmentation, their performances on CIFAR-100N largely outperforms all other methods. We also empirically test the performances of the above methods under the class-dependent noise settings which follow exactly the same $T$ as appeared in CIFAR-10N and CIFAR-100N, details are deferred to Table 9 in the Appendix.

Table 2: Comparison of test accuracies (%) on CIFAR-10N and CIFAR-100N (fine-label) using different methods. Top 3 performances are highlighted in **bold** (mean±standard deviation of 5 runs). More methods are included in the Appendix E as well as `http://noisylabels.com`.

| Method | CIFAR-10N | | | | | | CIFAR-100N | |
| --- | --- | --- | --- | --- | --- | --- | --- | --- |
| | Clean | Aggregate | Random 1 | Random 2 | Random 3 | Worst | Clean | Noisy |
| CE (Standard) | $92.92 \pm 0.11$ | $87.77 \pm 0.38$ | $85.02 \pm 0.65$ | $86.46 \pm 1.79$ | $85.16 \pm 0.61$ | $77.69 \pm 1.55$ | $76.70 \pm 0.74$ | $55.50 \pm 0.66$ |
| Forward $T$ (Patrini et al., 2017) | $93.02 \pm 0.12$ | $88.24 \pm 0.22$ | $86.88 \pm 0.50$ | $86.14 \pm 0.24$ | $87.04 \pm 0.35$ | $79.79 \pm 0.46$ | $76.18 \pm 0.37$ | $57.01 \pm 1.03$ |
| Co-teaching+ (Yu et al., 2019) | $92.41 \pm 0.20$ | $90.61 \pm 0.22$ | $89.70 \pm 0.27$ | $89.47 \pm 0.18$ | $89.54 \pm 0.22$ | $83.26 \pm 0.17$ | $70.99 \pm 0.22$ | $57.88 \pm 0.24$ |
| T-Revision (Xia et al., 2019) | $93.35 \pm 0.23$ | $88.52 \pm 0.17$ | $88.33 \pm 0.32$ | $87.71 \pm 1.02$ | $87.79 \pm 0.67$ | $80.48 \pm 1.20$ | $72.83 \pm 0.21$ | $51.55 \pm 0.31$ |
| Peer Loss (Liu & Guo, 2020) | $93.99 \pm 0.13$ | $90.75 \pm 0.25$ | $89.06 \pm 0.11$ | $88.76 \pm 0.19$ | $88.57 \pm 0.09$ | $82.00 \pm 0.60$ | $74.67 \pm 0.36$ | $57.59 \pm 0.61$ |
| ELR+ (Liu et al., 2020) | $\mathbf{95.39 \pm 0.05}$ | $\mathbf{94.83 \pm 0.10}$ | $94.43 \pm 0.41$ | $94.20 \pm 0.24$ | $94.34 \pm 0.22$ | $91.09 \pm 1.60$ | $\mathbf{78.57 \pm 0.12}$ | $\mathbf{66.72 \pm 0.07}$ |
| Positive-LS (Lukasik et al., 2020) | $94.77 \pm 0.17$ | $91.57 \pm 0.07$ | $89.80 \pm 0.28$ | $89.35 \pm 0.33$ | $89.82 \pm 0.14$ | $82.76 \pm 0.53$ | $76.25 \pm 0.35$ | $55.84 \pm 0.48$ |
| F-Div (Wei & Liu, 2020) | $94.88 \pm 0.12$ | $91.64 \pm 0.34$ | $89.70 \pm 0.40$ | $89.79 \pm 0.12$ | $89.55 \pm 0.49$ | $82.53 \pm 0.52$ | $76.14 \pm 0.36$ | $57.10 \pm 0.65$ |
| Divide-Mix (Li et al., 2020) | $\mathbf{95.37 \pm 0.14}$ | $\mathbf{95.01 \pm 0.71}$ | $\mathbf{95.16 \pm 0.19}$ | $\mathbf{95.23 \pm 0.07}$ | $\mathbf{95.21 \pm 0.14}$ | $\mathbf{92.56 \pm 0.42}$ | $76.94 \pm 0.22$ | $\mathbf{71.13 \pm 0.48}$ |
| Negative-LS (Wei et al., 2021) | $\mathbf{94.92 \pm 0.25}$ | $91.97 \pm 0.46$ | $90.29 \pm 0.32$ | $90.37 \pm 0.12$ | $90.13 \pm 0.19$ | $82.99 \pm 0.36$ | $\mathbf{77.06 \pm 0.73}$ | $58.59 \pm 0.98$ |
| CORES* (Cheng et al., 2021) | $94.16 \pm 0.11$ | $\mathbf{95.25 \pm 0.09}$ | $\mathbf{94.45 \pm 0.14}$ | $\mathbf{94.88 \pm 0.31}$ | $\mathbf{94.74 \pm 0.03}$ | $\mathbf{91.66 \pm 0.09}$ | $73.87 \pm 0.16$ | $55.72 \pm 0.42$ |
| VolMinNet (Li et al., 2021) | $92.14 \pm 0.30$ | $89.70 \pm 0.21$ | $88.30 \pm 0.12$ | $88.27 \pm 0.09$ | $88.19 \pm 0.41$ | $80.53 \pm 0.20$ | $70.61 \pm 0.88$ | $57.80 \pm 0.31$ |
| CAL (Zhu et al., 2021a) | $94.50 \pm 0.31$ | $91.97 \pm 0.32$ | $90.93 \pm 0.31$ | $90.75 \pm 0.30$ | $90.74 \pm 0.24$ | $85.36 \pm 0.16$ | $75.67 \pm 0.25$ | $61.73 \pm 0.42$ |
| PES (Semi) (Bai et al., 2021) | $94.76 \pm 0.20$ | $94.66 \pm 0.18$ | $\mathbf{95.06 \pm 0.15}$ | $\mathbf{95.19 \pm 0.23}$ | $\mathbf{95.22 \pm 0.13}$ | $\mathbf{92.68 \pm 0.22}$ | $\mathbf{77.92 \pm 0.04}$ | $\mathbf{70.36 \pm 0.33}$ |

**Observation 5: Performance gap (human noise v.s. synthetic noise)**  Continuing the reported observations in Section 4.1, in Table 3 we highlight that for most selected methods, class-dependent synthetic noise is much easier to learn on CIFAR-10, especially when the noise level is high. The gap is less obvious for CIFAR-100. However, we also observe that Divide-Mix (Li et al., 2020) fails to work well in low noise regime. Besides, ELR (Liu et al., 2020) performs even slight better when learning with real-world human noise than that on the synthetic class-dependent noise settings.

Table 3: Performance gap between human noise and class-dependent noise: test accuracy (trained on synthetic noise) - test accuracy (trained on human noise). Negative gaps are highlighted in red.

| Method | CIFAR-10 Gap | | | | | CIFAR-100 Gap |
|---|---|---|---|---|---|---|
| | Aggregate | Random 1 | Random 2 | Random 3 | Worst | Noisy |
| CE (Standard) | 4.35 | 6.01 | 4.50 | 5.82 | 9.00 | 1.20 |
| Forward $T$ (Patrini et al., 2017) | 4.30 | 4.82 | 4.86 | 4.27 | 7.08 | -0.14 |
| Co-teaching+ (Yu et al., 2019) | 0.89 | 0.92 | 0.86 | 1.05 | 2.63 | -0.61 |
| Peer Loss (Liu & Guo, 2020) | 1.90 | 2.42 | 2.74 | 1.95 | 4.67 | -0.85 |
| ELR (Liu et al., 2020) | -0.78 | -0.81 | -0.97 | -0.51 | -1.64 | 1.05 |
| F-Div (Wei & Liu, 2020) | 0.72 | 1.62 | 1.33 | 1.65 | 4.14 | 1.31 |
| Divide-Mix (Li et al., 2020) | -0.01 | 0.44 | 0.42 | 0.28 | 0.29 | 0.65 |
| Negative-LS (Wei et al., 2021) | 0.77 | 1.31 | 1.08 | 1.36 | 4.00 | 1.26 |
| JoCoR (Wei et al., 2020) | 0.35 | 0.78 | 0.68 | 1.01 | 2.43 | -0.48 |
| CORES$^2$ (Cheng et al., 2021) | 1.49 | 1.69 | 1.52 | 1.66 | 1.67 | -0.72 |
| CAL (Zhu et al., 2021a) | 0.25 | 0.04 | 0.04 | 0.09 | 0.44 | 0.47 |

## 5.2 MEMORIZATION EFFECTS

When learning with noisy labels on CIFAR-10 and CIFAR-100 datasets, empirical observations (Arpit et al., 2017; Liu et al., 2020; Xia et al., 2020a; Zhang et al., 2021a) on synthetic noise settings suggest that deep neural networks firstly fit on samples with clean labels, then gradually over-fit and memorize samples with wrong/noisy labels (Xie et al., 2021). We next explore the memorization of clean and noisy labels on CIFAR-10N and CIFAR-100N.

**Definition 2 (Memorized feature)** *In a K-class classification task, given a trained classifier $f$, a feature $x$ and confidence threshold $\eta$, $x$ is memorized by $f$ if $\exists i \in [K]$ s.t. $\mathbb{P}(f(x) = i) > \eta$.*

In Figure 6, we train CE loss with a ResNet-34 (He et al., 2016) neural network on three noisy label sets of CIFAR-10N: aggre-label (left column), random-label1 (middle column) and worst-label (right column). While visualizing the memorization ($\eta = 0.95$) on training samples, we split the train data into two parts: images with clean labels (the annotation matches the clean label) and wrong labels (the rest). We observe that: *deep neural nets memorize features more easily when learning with real-world human annotations than synthetic ones.* This is attributed to the fact that, compared with synthetic label noise, human annotators are prone to providing wrong labels on more misleading/ambiguous images or complex patterns. Given the same noise level, learning with human noise labels is more challenging and deep neural nets over-fit on features of wrong annotations inevitably.

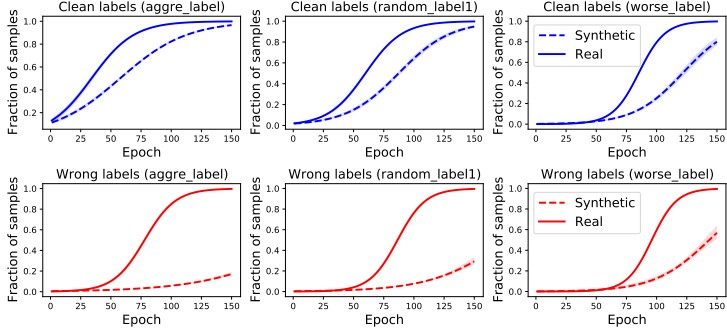

Figure 6: Memorization of clean/correct and wrong labels on CIFAR-10N and synthetic noise with same $T$: red line denotes the percentage of memorized (wrongly predicted) samples, blue line denotes that of correctly predicted ones.

## 6 CONCLUSIONS

Building upon CIFAR datasets, we provide the weakly supervised learning community with two accessible and easy-to-use benchmarks: CIFAR-10N and CIFAR-100N. We introduce new observations from human annotations such as imbalanced annotations, the flipping of noisy labels among similar features, co-existing labels in CIFAR-100N, etc. From the perspective of noise transitions, we qualitatively show that human noise is indeed feature-dependent and differs substantially from synthetic class-dependent label noise using hypothesis testing. We empirically compare the robustness of a large quantity of popular methods when learning with CIFAR-10N, CIFAR-100N and synthetic noisy CIFAR datasets. We also consistently observe the large performance gap between human noise and synthetic noise, as well as the different memorization behavior on training samples.

ACKNOWLEDGEMENT

This work is supported by a University of California, Santa Cruz startup fund, the National Science Foundation (NSF) under grant IIS-2007951 and IIS-2143895. TL was partially supported by Australian Research Council Projects DE-190101473 and DP-220102121.

LICENSE

The released datasets CIFAR-N are publicly available at `http://noisylabels.com`, under the Attribution-NonCommercial 4.0 International (CC BY-NC 4.0) license. You are free to share and adapt, while under the terms of attribution and non-commercial use.

OPEN COMPETITION

Moving forward, we would like to invite researchers to openly compete using our CIFAR-N datasets to advance the field. Correspondingly, we will actively update our leaderboards. Featured competitions will include building better and more accurate models, estimating noise transition matrix, and detecting corrupted labels. This year (2022), we will host the inaugural public CIFAR-N competition with IJCAI-ECAI 2022. We will be actively maintaining `http://noisylabels.com` to disseminate future information.

ETHICS STATEMENT

This paper does not raise any ethics concerns. Our work contains the human subject study which involves only simply image annotation tasks in Amazon Mechanical Turk. The study has been carefully reviewed and received Institutional Review Board (IRB) exempt approval. We have followed the outlined protocol to perform the data collection. Our implemented human subject studies:

- **Do not** have negative consequences, i.e., leaking the private information that would identify human annotators on Amazon Mechanical Turk.
- **Do not** misrepresent work that might be competing or related.
- **Do not** present misleading insights. Our work does not present applications that can lead to misuse.
- **Do not** introduce bias or fairness concerns, and research integrity issues.
- We make the collected datasets (CIFAR-N) and the leaderboard publicly available at `http://noisylabels.com`. A starter code is provided in `https://github.com/UCSC-REAL/cifar-10-100n`.

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

# APPENDIX

The Appendix is organized as follows:

- Section A: the details of dataset collection, processing and information of CIFAR-10N.
- Section B: the details of dataset collection, processing and information of CIFAR-100N.
- Section C: the comparison of label collection procedure among CIFAR, CIFAR-10H, and CIFAR-N.
- Section D: more detailed hypothesis testing results of CIFAR-N.
- Section E: additional experiment results and details.

Before proceeding to the appendix, we want to highlight our boarder impacts as follows.

**Broader Impacts**  Our observations and contributions may have following potential broader impacts:

- **Crowd-sourcing of data annotations in computer vision:** CIFAR-N may be further used for studying/proposing simulations of human annotations in crowd-sourcing, where the expenses of obtaining human annotations are often tremendous.
- **A template for hypothesizing label noise patterns:** Our hypothesis testing method of instance-dependent label noise may provide a quantitative tool for testing the simulated human labels.
- **Benchmarking effort is important:** Although learning from noisy labels has witnessed thriving developments, we often observed conflicting comparisons due to the randomness in the synthetic noisy labels. While there exist several datasets with real human noise, we view our contribution as complementary to existing ones, due to the elaborated reasons above. We believe benchmarking existing and population solutions is an important technical contribution to the community.
- **Understanding real-world label noise:** Our observations and the provided human-annotated labels help with understanding real-world label noise. Besides, our observations of the **multi-label** issues in CIFAR-100 impose a new label noise pattern that is largely neglected.
- **Motivations for real-world label noise solutions:** our observations, especially the memorizing effects of real-world label noise may provide the literature with motivations for addressing real-world label noise.

## A  CIFAR-10N REAL-WORLD NOISY LABEL BENCHMARK

In this section, we introduce the preparation for data collection, collection procedure of CIFAR-10N, the workers' behaviors, and more detailed statistics of the obtained labels.

### A.1  CASE STUDIES BEFORE THE FORMAL COLLECTION

To make the collection procedure reasonable and efficient, we firstly upload a few batches (500 images / batch) to test the behaviors of workers. Our observations show that the workers on image classification tasks may possibly incur following phenomenons:

- **Bots:** with the appearance of bots, the accepted HIT may result in low-quality or meaningless responses if the bot is able to pick answers and maliciously/randomly submit them. Otherwise, the bot accepts the HIT but could not submit annotations. The accepted HIT may have to be re-assigned to another work after this HIT becomes expired which results in inefficient data collection.
- **A large variance of the workers' contribution:** empirical observations show that a large amount of workers contribute too few HITs, while some professional workers upload with much less time. Thus, there exists a large variance in the number of the worker's submitted HITs and the noise label pattern is substantially controlled by only a few workers.
- **Incomplete submission:** when there are more than one image per HIT, a worker would possibly neglect to click the label of one or more images, for example, the worker may only choose easy tasks to annotate and skip tough ones. The incomplete submission complexes the reassignment

of images with missing annotations, the processing of annotated results as well as the individual payment.

## A.2 DATASET COLLECTION

To alleviate the impacts of aforementioned phenomenons, we randomly split the training dataset of CIFAR-10 without replacement into ten batches. Each batch contains 500 HITs. To ensure most workers do not contribute too few to the annotation task, we include ten $32 \times 32$ images per HIT. Each HIT is then randomly assigned to three independent workers. The worker gains a base reward $0.03 after submitting all annotations of one HIT within 2 minutes. The worker can not submit the annotations unless all appeared images in the assigned HIT are labeled. The averaged number of submitted HITs per worker is 7. Workers with no less 7 submissions share $200 bonus rewards. Note that we constrain the time duration for each assignment and re-design the interface, bots are less likely to finish our task either on time or under the procedure. What is more, we didn't make use of any ground-truth clean labels to approve or reject submissions. We only block and reject workers who submit answers with fixed/regular distribution patterns.

## A.3 THE WORKERS' BEHAVIORS

There are 747 independent workers contribute to the construction of CIFAR-10N. As depicted in Figure 7a, most workers can submit the labels for ten images within one minute. Although we do observe that a small amount of assignments ($\leq 0.2\%$) are finished within 10 seconds, which are likely to be low-quality responses. The work time in seconds have the mean 46.7, the standard deviation 21.2 and the interquartile (25th percentile — 75th percentile) range is $[31, 58]$. Among these 747 workers, most of them annotated more than 80 images. The number of annotated images per work has the mean 201, the standard deviation 329 and the interquartile range is $[30, 220]$.

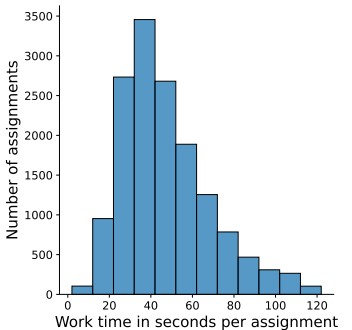
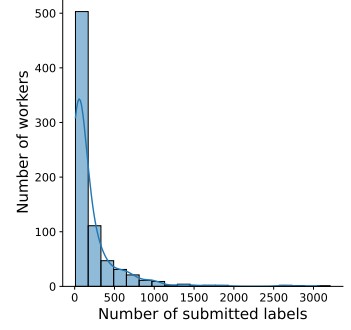

(a) Distribution of work time in seconds per HIT.

(b) Distribution of the amount of submitted labels.

Figure 7: The behaviors of workers in the collection of CIFAR-10N.

## A.4 MORE DETAILED DATASET STATISTICS

Table 4 includes the summarized statistics of CIFAR-10N for each batch. In Table 4, the statistics "Consensus" means the three labelers have a consensus on the label of the same image. We do observe that several batches tend out to be more challenging for human workers to annotate, i.e, the noise rate appeared in Batch3 is clearly higher than those of Batch4 and Batch5. The difference of noise rate is especially significant on the "Worst" label set. We conjecture that there might exist more human annotators who malicious submit low-quality annotations when working on Batch3.

Note that while the number of images are the same for each clean label, across all the five noisy label sets of CIFAR-10N, we observe that human annotators have different preferences for similar classes, i.e, they are more likely to annotate an image to be an automobile rather than the truck, to be the horse rather than the deer (see Figure 8). The aggregated labels appear more frequently in automobile and ship, and less frequently in deer and cat. This gap of frequency becomes more clear in the worst label set.

Table 4: Consensus and noise levels (%) of each noisy label set in 10 batches (CIFAR-10N).

| Statistics | Batch1 | Batch2 | Batch3 | Batch4 | Batch5 | Batch6 | Batch7 | Batch8 | Batch9 | Batch10 | Overall |
|---|---|---|---|---|---|---|---|---|---|---|---|
| **Consensus** | 53.32 | 64.26 | 40.20 | 68.42 | 70.04 | 62.90 | 58.10 | 66.98 | 58.00 | 60.52 | 60.27 |
| **Aggregate** | 10.20 | 7.92 | 10.76 | 7.90 | 7.14 | 7.72 | 10.12 | 8.42 | 10.80 | 9.30 | 9.03 |
| **Random 1** | 20.40 | 15.22 | 23.74 | 14.00 | 13.44 | 15.56 | 18.92 | 14.96 | 18.36 | 17.74 | 17.23 |
| **Random 2** | 21.54 | 15.36 | 28.20 | 14.80 | 12.92 | 16.08 | 18.70 | 15.18 | 20.74 | 17.70 | 18.12 |
| **Random 3** | 19.24 | 17.54 | 23.24 | 13.84 | 13.62 | 16.84 | 18.34 | 15.46 | 20.24 | 18.04 | 17.64 |
| **Worst** | 47.06 | 36.40 | 59.44 | 32.20 | 30.44 | 37.50 | 42.54 | 33.56 | 42.88 | 40.06 | 40.21 |

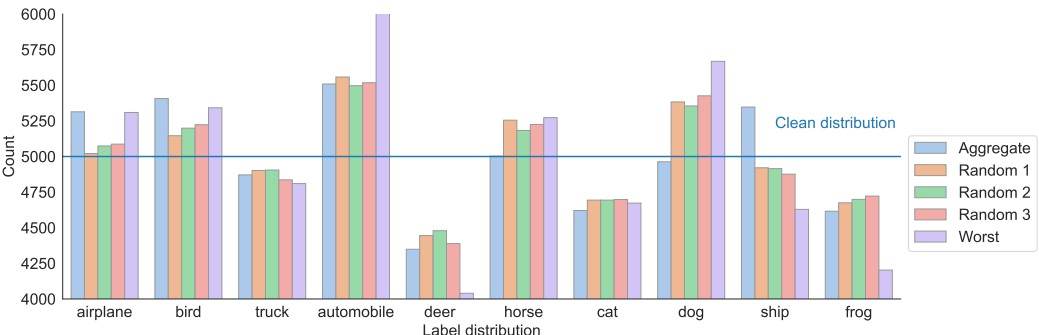

Figure 8: Categorical distribution of noisy label sets in CIFAR-10N.

# B    CIFAR-100N REAL-WORLD NOISY LABEL BENCHMARK

In this section, we introduce the preparation for data collection, collection procedure of CIFAR-100N, and more detailed statistics of the obtained labels.

## B.1    CASE STUDIES BEFORE THE FORMAL COLLECTION

To make the collection procedure reasonable and efficient, we firstly upload a few batches (500 images / batch) to test the behaviors of workers. Our observations show that the workers on CIFAR-100 classification tasks may not only have mentioned phenomenons in A, but have following issues as well:

- **Hard and time consuming:** directly let workers to find the best matched label for each image from 100 possible classes is time assuming. Our case study shows that finding the label per image takes averaged 5-6 minutes. The work load as well as the difficulty level stop many workers from contributing two or more submissions.

- **Lack of background knowledge:** distinguishing several classes require some preliminary knowledge in biology, for example, it is common for workers to select a wrong super-class, especially for animal related ones: "aquatic mammals" and "fish". And the differences among some fine labels are hard to recognize, i.e, "trees" (oak, palm, pine), "medium-sized mammals" (porcupine, possum, raccoon, skunk), etc.

- **Label aggregation has few effects:** our empirical observations show that the the decrease of noise rates from aggregated labels given by 3 independent workers is less significant than the results on CIFAR-10N.

## B.2    DATASET COLLECTION

To deal with above mentioned issues, we firstly split the training dataset of CIFAR-100 without replacement into ten batches. Each batch contains 1000 HITs. We include five $96 \times 96$ images per HIT which are reshaped from $32 \times 32$ ones in CIFAR-100 train images. Only one worker is assigned with each HIT. Instead of requesting workers to find the best matched label from 100 labels directly, we group the 100 classes into 20 super-classes which are slightly different from the 20 raw "coarse" categories given by Krizhevsky et al. (2009). The 20 newly defined super-classes are summarized in Table 5. And the new division in each super-class is stated in Table 6. In order to reduce the workload of workers, they are instructed to firstly select the super-class for each image.

They will then re-directed to the corresponding 4-6 fine labels. Note that some super-classes are hard to recognize without some prior knowledge in biology, we provide workers with easy access to re-select the super-class for the current image, and every fine label has an example image from Google images for references.. If the worker luckily finds the most suitable fine label from her point of view, we also supply the jumping button so that the worker efficiently goes to the next image or the final submission of this HIT. A worker gains base reward $0.07 after submitting the answers of each HIT within 6 minutes (averaged working time is less than 90 seconds). Similar to the setting in the collection of CIFAR-10N, huge bonus applied to workers who contribute more HITs than the averaged number of submissions. Workers who submit answers with fixed/regular distribution patterns will be blocked and rejected all submitted results.

Table 5: Newly defined super-classes in CIFAR-100N.

| (1) Aquatic mammals | (2) Fish | (3) Flowers |
|---|---|---|
| (4) Food containers | (5) Fruit, vegetables and mushrooms | (6) Household electrical devices |
| (7) Household furniture | (8) Insects | (9) Large carnivores and bear |
| (10) Large man-made outdoor things | (11) Large natural outdoor scenes | (12) Large omnivores and herbivores |
| (13) Medium-sized mammals | (14) Non-insect invertebrates | (15) People |
| (16) Reptiles | (17) Small mammals | (18) Trees |
| (19) Transportation vehicles | (20) Other vehicles | |

Table 6: Division of each super-class in CIFAR-100N.

| Super-class | Fine-class |
|---|---|
| Aquatic mammals | beaver, dolphin, otter, seal, whale |
| Fish | aquarium fish, flatfish, ray, shark, trout |
| Flowers | orchids, poppies, roses, sunflowers, tulips |
| Food containers | bottles, bowls, cans, cups, plates |
| Fruit, vegetables and mushrooms | apples, mushrooms, oranges, pears, sweet peppers |
| Household electrical devices | clock, computer keyboard, lamp, telephone, television |
| Household furniture | bed, chair, couch, table, wardrobe |
| Insects | bee, beetle, butterfly, caterpillar, cockroach |
| Large carnivores and bear | bear, leopard, lion, tiger, wolf |
| Large man-made outdoor things | bridge, castle, house, road, skyscraper |
| Large natural outdoor scenes | cloud, forest, mountain, plain, sea |
| Large omnivores and herbivores | camel, cattle, chimpanzee, elephant, kangaroo |
| Medium-sized mammals | fox, porcupine, possum, raccoon, skunk |
| Non-insect invertebrates | crab, lobster, snail, spider, worm |
| People | baby, boy, girl, man, woman |
| Reptiles | crocodile, dinosaur, lizard, snake, turtle |
| Small mammals | hamster, mouse, rabbit, shrew, squirrel |
| Trees | maple, oak, palm, pine, willow |
| Transportation vehicles | bicycle, bus, motorcycle, pickup truck, train, streetcar |
| Other vehicles | lawn-mower, rocket, tank, tractor |

## B.3 MORE DETAILED DATASET STATISTICS

For CIFAR-100N dataset, each image contains a coarse label and a fine label given by a human annotator. Most batches have approximately 40% noisy fine labels and 25 % noisy coarse labels. The overall noise level of coarse and fine labels are 25.60% and 40.20%, respectively. A detailed summary of noise level is available in Table 7 which covers the statistics of each batch. Human annotators annotate frequently on classes which are outside of the clean-coarse, i.e., 25% noisy labels fall outside of the super-class and 15% inside the super-class.

Table 7: Noise level (%) on CIFAR-100N.

| Statistics | Batch1 | Batch2 | Batch3 | Batch4 | Batch5 | Batch6 | Batch7 | Batch8 | batch9 | Batch10 | Overall |
|---|---|---|---|---|---|---|---|---|---|---|---|
| **100-class** | 40.30 | 40.76 | 40.84 | 40.16 | 42.50 | 34.44 | 43.26 | 40.34 | 43.66 | 35.84 | 40.20 |
| **20-class** | 26.82 | 28.02 | 25.54 | 25.76 | 27.02 | 20.80 | 28.62 | 25.56 | 26.92 | 20.98 | 25.60 |

**Imbalanced annotations.** The phenomenon of imbalanced annotations also appears substantially as shown in Figure 9, which presents the distribution of noisy labels for each fine class. "Man" appears $\geq 750$ times, while "Streetcar" only has $\approx 200$ annotations.

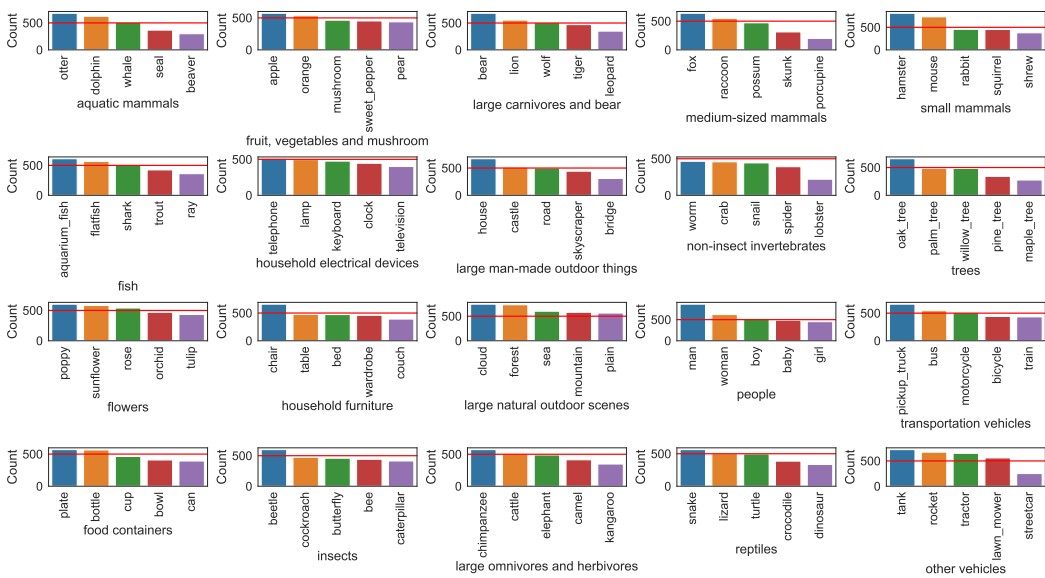

Figure 9: Categorical distribution of noisy labels on CIFAR-100N: the red line in each subplot indicates that the number of each clean fine class is 500.

**Noisy label flips to similar features.** In CIFAR-100N, most fine classes are more likely to be mislabeled into less than four fine classes. In Figure 10, we show top three wrongly annotated fine labels for several fine classes that have a relative large noise rate. Due to the low-resolution of images, a number of noisy labels are annotated in pair of classes, i.e, $\approx 20\%$ of "snake" and "worm" images are mislabeled between each other, similarly for "cockroack"-"beetle", "fox"-"wolve", etc. While some other noisy labels are more frequently annotated within more classes, such as "boy"-"baby"-"girl"-"man", "shark"-"whale"-"dolphin"-"trout", etc, which share similar features.

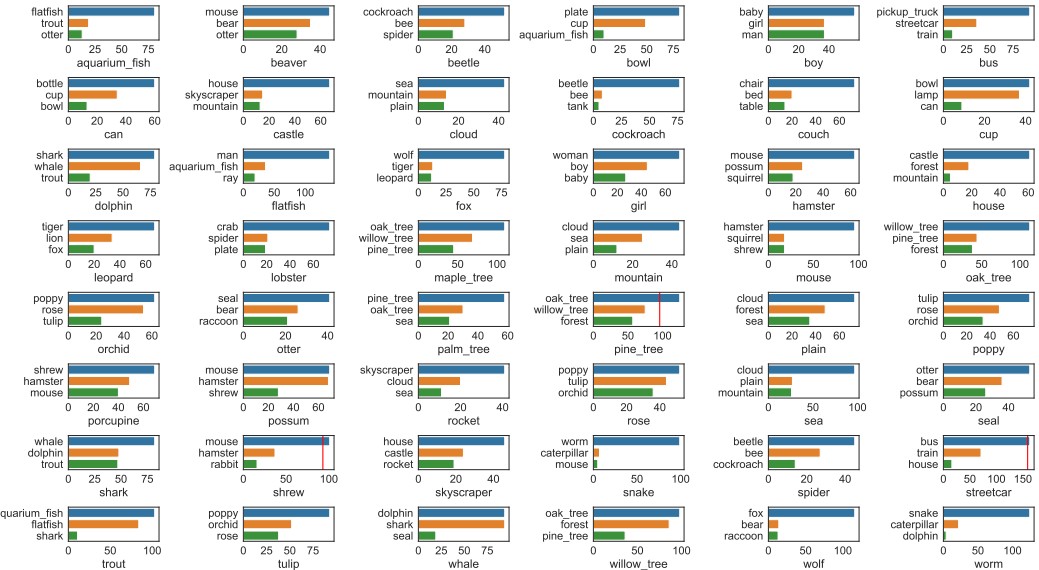

Figure 10: Top 3 wrongly annotated fine labels for each selected fine classes. For "pine tree", "shrew", "streetcar", the dominant class is the **wrong** class. The corresponding number of correct annotations are highlighted with red lines.

**Transition matrices of CIFAR-100N**   In the class-dependent label noise setting, suppose the label noise is conditional independent of the feature, the noise transition matrices of CIFAR-100N are best described by a mixture of symmetric and asymmetric $T$. For CIFAR-100N, we heatmap the coarse and fine noisy labels w.r.t. the clean labels. In Figure 11, the clean label flips into one or more similar classes more often. And the remaining classes follow the symmetric noise model with a low noise rate. Apparently, current synthetic class-dependent noisy settings are not as complex as the real-world human annotated label noise.

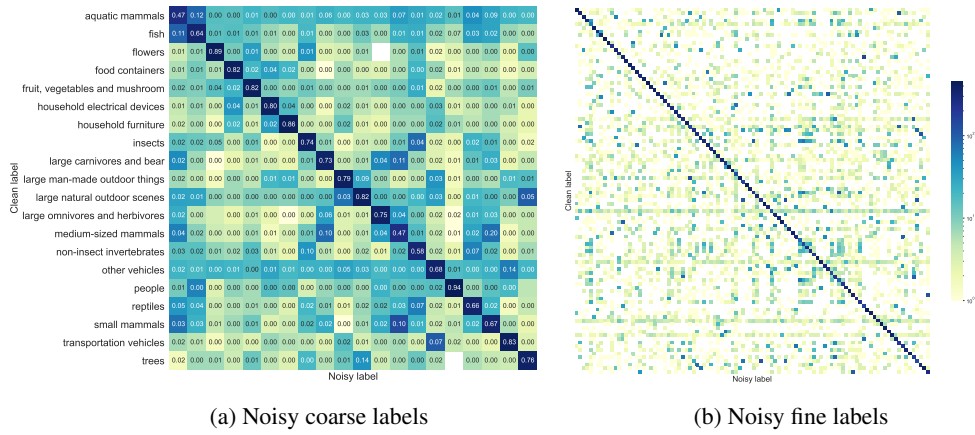

(a) Noisy coarse labels                    (b) Noisy fine labels

Figure 11: Transition matrix of CIFAR-100h noisy labels: coarse labels (left) and fine labels (right).

## C   COMPARISONS OF THE LABEL COLLECTION PROCEDURE

### C.1   COMPARISONS BETWEEN CIFAR AND CIFAR-N

We summarize the label collection procedure of CIFAR dataset as below:

- **Images sources:** the images of CIFAR come from down-scaled images from 80 million color images obtained by various search engines. The corresponding searching terms are viewed as a noisy source label/class;
- **Remove irrelevant images:** the designer provided each student (labeller) with a class. Students were asked to verify all the images which were found with that class as the search term. And reject images that are outside of the assigned class.
- **Payment:** the payment for each student/labeller per hour is fixed.
- **Noise rate:** with additional personal verification, the noise rate of each class is negligible, especially for CIFAR-10.

We highlight the differences of label collection procedure between CIFAR and CIFAR-N as below:

- **Annotation method:** CIFAR adopted search engines such that they obtained images by specifying the class name (conditioned on the label, from label to feature); while CIFAR-N makes use of CIFAR images and pays workers for annotating labels (from feature to label).
- **Payment and incentives:** CIFAR fixed the payment for student workers (self-collected), there is no incentive to rush. As for CIFAR-N, we did not have any constraints on the workers via Amazon Mechanical Turk, i.e., the salary, the education level, etc. Besides, we set time limits and bonuses for each annotation task, there are incentives to rush and meanwhile provide high-quality labels.
- **Noise rate:** for CIFAR, the noise rate can be viewed as 0. While for CIFAR-N, we provide several noisy label sets which are of various noise rates.

## C.2   COMPARISONS BETWEEN CIFAR-10H AND CIFAR-N

We firstly summarize the main collection procedure of CIFAR-10H, and then move to discuss the difference in the label collection procedure.

The label collection procedure of CIFAR-10H dataset is given below:

- **No time limit:** participants were asked to select the label of each image from the surrounding text with no time limit.

- **Controlled workload:** Each participant is assigned 200 images (20 per class);

- **Attention check (intervention):** Participants with a low accuracy (<75%) on (easy-to-recognize) images were removed.

- **Other minor differences:** Label positions are shuffled among participants;47-63 annotations per image; payment: 0.0075/10 images.

We want to highlight that the central query for CIFAR-10H is to understand the benefits of uncertainty in human annotations to improve the generalization power of the trained model. Therefore, a number of controls and interventions were applied when building CIFAR-10H to control the human noise rates to be not excessive to disturb the above benefit study (details below). We believe this aspect renders the dataset not super appropriate for the relevant studies reported in Section 4 and 5. Our detailed reasons come as follows:

- **Intervened real-world human noise pattern**
  - **Purpose of the dataset construction:** CIFAR-10H targets to identify the benefits from increasing the richness of label distributions (hard label $\rightarrow$ soft label) for image classification tasks. The soft labels are constructed by human uncertainty. *CIFAR-10H may not fully reveal the real-world human annotation noise due to the check and removal procedure as we described above in attention check.* While we aim to study the real-world label noise pattern: we only reject uninformative and spamming annotation patterns (e.g., labeling every task as class 1) and we do not restrict the number of annotations required from different workers with different working efficiency. We accept submissions even if a worker has a moderate accuracy (e.g., <60%) and meanwhile reward workers that contribute a large number of annotations.

  - **Noise rate:** we randomly select the i-th (e.g., 1,2,...,10-th) annotated label for each test image in CIFAR-10-H, and there are approximately 5% wrong labels in the annotation. In CIFAR-10N, the random noise rate is around 18%. For CIFAR-100N, the noise rate increased to 40.20%. *CIFAR-10H has a much smaller noise rate due to the control intervention, and due to different objects of the collection.* We believe that a very low noise rate may deviate from real-world human noise.

  Therefore, we think our collection might have better captured the real-world noise patterns.

- **Training data V.S. Test data**
  - Training on CIFAR-10 test data may *lead to a model performance drop.* It is reported that, when trained using the much smaller test data, the generalization accuracy on training data is only about 83% (Peterson et al., 2019). Note the standard training and testing on CIFAR-10 has an accuracy of about 93%. With added label noise, the substantial drop of the number of training data limits the possibility of fully evaluating the potentials and properties of the competing benchmark methods (e.g., learning and showing some of the established theoretical properties of a particular method might require a sufficient number of training data).

  - Looking forward, we think it might be beneficial to let the learning-with-noisy-label community have an option of training using 50k training data and testing on the 10k test data, the same and standard way as other learning communities have developed and evaluated algorithms using CIFAR-10 data. As we benchmarked in Table 2, most of the existing works are tuned (e.g., pre-trained models for representation extraction for CIFAR-10, etc) for the training with 50k training images. This can help the community better align and calibrate the progress, as compared to other learning tasks (e.g., supervised learning, semi-supervised learning, etc).

# D    HYPOTHESIS TESTING OF CIFAR-N

## D.1    HYPOTHESIS TESTING OF CIFAR-10N

In this section, we include the hypothesis testing results for each class as. As described in Figure 12, the $p$-value of the human noise label w.r.t each clean class (except for "automobile") in CIFAR-10N is less than $0.05$ (*). Most of the classes (except for three of them) achieve a $p$-value that is smaller than $0.01$ (***). Since there exists classes in CIFAR-10N such that the null hypothesis is rejected with the significance value $\alpha$, hypothesis "**human annotated label noise in CIFAR-10N is feature-dependent**" is accepted.

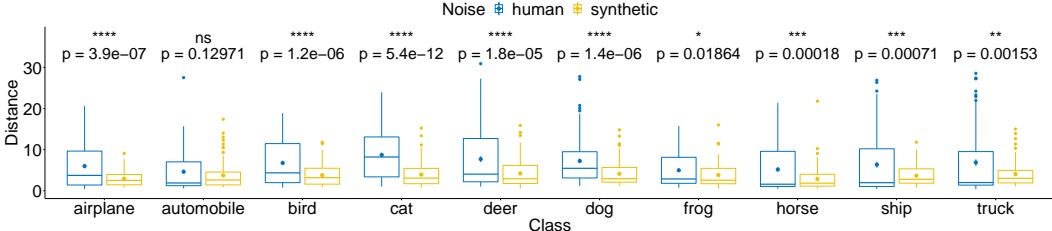

Figure 12: Hypothesis testing results: we adopt two samples $\left(\{d_{i,\nu}^{(1)}\}_{i\in[K],\nu\in[5]}\right.$ and $\left.\{d_{i,\nu}^{(2)}\}_{i\in[K],\nu\in[5]}\right)$ student t-tests. The $p$-value and the significance level are shown for each class. Significance levels are denoted as 'ns': $p > 0.05$; '*': $p \leq 0.05$; '**': $p \leq 0.01$; '***': $p \leq 0.001$; '****': $p \leq 0.0001$. When removed the class constraint, we obtained $p = 1.8e^{-36}$ for the whole data.

## D.2    HYPOTHESIS TESTING OF CIFAR-100N

We further statistically test whether the human-noise is feature-dependent or not in CIFAR-100N. The null hypothesis $\mathbf{H_0}$ and the corresponding alternate hypothesis $\mathbf{H_1}$ are defined as:

$\mathbf{H_0}$ : Human annotated label noise in CIFAR-100N is *feature-independent*;

$\mathbf{H_1}$ : Human annotated label noise in CIFAR-100N is *feature-dependent*.

Note that the synthetic label noise is supposed to be feature-independent, the above hypotheses are converted to:

$\mathbf{H_0}$ : Human annotated label noise in CIFAR-100N is *the same as* the corresponding synthetic one;

$\mathbf{H_1}$ : Human annotated label noise in CIFAR-100N is *different from* the corresponding synthetic one.

As implemented for CIFAR-10N, we adopt the distance between transition vectors $\boldsymbol{p}_{i,\nu}$ across different noise as measure of the difference between human noise and synthetic noise, e.g., $d_{i,\nu}^{(1)} := \|\boldsymbol{p}_{i,\nu}^{\text{human}} - \boldsymbol{p}_{i,\nu}^{\text{synthetic}}\|_2^2$. As contrast, we need to compare $d_{i,\nu}^{(1)}$ with $d_{i,\nu}^{(2)} := \|\boldsymbol{p}_{i,\nu}^{\text{synthetic}'} - \boldsymbol{p}_{i,\nu}^{\text{synthetic}}\|_2^2$, where $\boldsymbol{p}_{i,\nu}^{\text{synthetic}'}$ denotes the transition vector from the same synthetic noise but different clustering result (caused by random data augmentation). Intuitively, if $d_{i,\nu}^{(1)}$ is much greater than $d_{i,\nu}^{(2)}$, we should accept $\mathbf{H_1}$. The above hypotheses are then equivalent to:

$\mathbf{H_0}$ : $\{d_{i,\nu}^{(1)}\}_{i\in[K],\nu\in[5]}$ come from the *same* distribution as $\{d_{i,\nu}^{(2)}\}_{i\in[K],\nu\in[5]}$;

$\mathbf{H_1}$ : $\{d_{i,\nu}^{(1)}\}_{i\in[K],\nu\in[5]}$ come from *different* distributions from $\{d_{i,\nu}^{(2)}\}_{i\in[K],\nu\in[5]}$.

We repeat the generation of $d_{i,\nu}$ for 10 times, where the images are modified with different data augmentations each time. We choose the significance level $\alpha = 0.05$ and perform a two-sided t-test w.r.t $\{d_{i,\nu}^{(1)}\}_{i\in[K],\nu\in[5]}$ and $\{d_{i,\nu}^{(2)}\}_{i\in[K],\nu\in[5]}$. As described in Figure 13, the $p$-value of approximately 50 classes is less than $\alpha = 0.05$. Thus, the null hypothesis for the human noisy labels in CIFAR-100N is rejected with the significance value $\alpha$. And we accept the hypothesis:

**Human annotated label noise in CIFAR-100N is *feature-dependent*.**

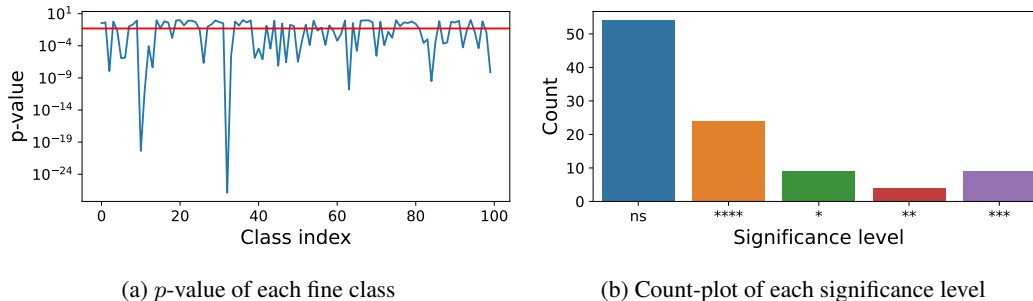

(a) *p*-value of each fine class        (b) Count-plot of each significance level

Figure 13: Hypothesis testing results of CIFAR-100N: we adopt two samples $\big(\{d_{i,\nu}^{(1)}\}_{i\in[K],\nu\in[5]}$ and $\{d_{i,\nu}^{(2)}\}_{i\in[K],\nu\in[5]}\big)$ student t-tests. The (a) *p*-value and the (b) significance level are shown for each class. Significance levels are denoted as 'ns': $p > 0.05$; '*': $p \leq 0.05$; '**': $p \leq 0.01$; '***': $p \leq 0.001$; '****': $p \leq 0.0001$. When removed the class constraint, $p = 5.2e^{-16}$ for the whole data.

**Beyond feature dependency**  As shown in Figure 13 (b), although the overall noisy fine labels in CIFAR-100N are feature dependent, the noise transition vectors of around 50 classes can indeed be viewed as class dependent by referring to their significance levels. Thus, we can conclude that real-world human annotated noisy labels may be feature independent (class dependent) for certain classes.

# E  ADDITIONAL EXPERIMENT RESULTS

## E.1  PERFORMANCE COMPARISONS ON CIFAR-N DATASETS

We include a larger family of robust methods when learning with CIFAR-N in the Table 8. Note that both ELR+ (Liu et al., 2020) and Divide-Mix (Li et al., 2020) adopt two networks with advanced strategies such as mix-up data augmentation, their performances on CIFAR-100N largely outperforms all other methods. The performance gap between ELR and ELR+ becomes much larger when the noise level is high.

Table 8: Comparison of test accuracies (%) on CIFAR-10N and CIFAR-100N (fine-label) using different methods. Top 3 performances are highlighted in **bold** (mean±standard deviation of 5 runs). SOP (Liu et al., 2022) trained on Pre-act ResNet-18 architecture, while all other methods are reproduced on ResNet-34.

| Method | CIFAR-10N | | | | | | CIFAR-100N | |
|---|---|---|---|---|---|---|---|---|
| | Clean | Aggregate | Random 1 | Random 2 | Random 3 | Worst | Clean | Noisy |
| CE (Standard) | 92.92 ± 0.11 | 87.77 ± 0.38 | 85.02 ± 0.65 | 86.46 ± 1.79 | 85.16 ± 0.61 | 77.69 ± 1.55 | 76.70 ± 0.74 | 55.50 ± 0.66 |
| Forward $T$ (Patrini et al., 2017) | 93.02 ± 0.12 | 88.24 ± 0.22 | 86.88 ± 0.50 | 86.14 ± 0.24 | 87.04 ± 0.35 | 79.79 ± 0.46 | 76.18 ± 0.37 | 57.01 ± 1.03 |
| Backward $T$ (Patrini et al., 2017) | 93.10 ± 0.05 | 88.13 ± 0.29 | 87.14 ± 0.34 | 86.28 ± 0.80 | 86.86 ± 0.41 | 77.61 ± 1.05 | 76.79 ± 0.60 | 57.14 ± 0.92 |
| GCE (Zhang & Sabuncu, 2018) | 92.83 ± 0.16 | 87.85 ± 0.70 | 87.61 ± 0.28 | 87.70 ± 0.56 | 87.58 ± 0.29 | 80.66 ± 0.35 | 76.35 ± 0.48 | 56.73 ± 0.30 |
| Co-teaching (Han et al., 2018) | 93.35 ± 0.14 | 91.20 ± 0.13 | 90.33 ± 0.13 | 90.30 ± 0.17 | 90.15 ± 0.18 | 83.83 ± 0.13 | 73.46 ± 0.09 | 60.37 ± 0.27 |
| Co-teaching+ (Yu et al., 2019) | 92.41 ± 0.20 | 90.61 ± 0.22 | 89.70 ± 0.27 | 89.47 ± 0.18 | 89.54 ± 0.22 | 83.26 ± 0.17 | 70.99 ± 0.22 | 57.88 ± 0.24 |
| T-Revision (Xia et al., 2019) | 93.35 ± 0.23 | 88.52 ± 0.17 | 88.33 ± 0.32 | 87.71 ± 1.02 | 87.79 ± 0.67 | 80.48 ± 1.20 | 72.83 ± 0.21 | 51.55 ± 0.31 |
| Peer Loss (Liu & Guo, 2020) | 93.99 ± 0.13 | 90.75 ± 0.25 | 89.06 ± 0.11 | 88.76 ± 0.19 | 88.57 ± 0.09 | 82.00 ± 0.60 | 74.67 ± 0.36 | 57.59 ± 0.61 |
| ELR (Liu et al., 2020) | 93.45 ± 0.65 | 92.38 ± 0.64 | 91.46 ± 0.38 | 91.61 ± 0.16 | 91.41 ± 0.44 | 83.58 ± 1.13 | 72.78 ± 0.80 | 58.94 ± 0.92 |
| ELR+ (Liu et al., 2020) | **95.39 ± 0.05** | 94.83 ± 0.10 | 94.43 ± 0.41 | 94.20 ± 0.24 | 94.34 ± 0.22 | 91.09 ± 1.60 | **78.57 ± 0.12** | 66.72 ± 0.07 |
| Positive-LS (Lukasik et al., 2020) | 94.77 ± 0.17 | 91.57 ± 0.07 | 89.80 ± 0.28 | 89.35 ± 0.33 | 89.82 ± 0.14 | 82.76 ± 0.53 | 76.25 ± 0.35 | 55.84 ± 0.48 |
| F-Div (Wei & Liu, 2020) | 94.88 ± 0.12 | 91.64 ± 0.34 | 89.70 ± 0.40 | 89.79 ± 0.12 | 89.55 ± 0.49 | 82.53 ± 0.52 | 76.14 ± 0.36 | 57.10 ± 0.65 |
| Divide-Mix (Li et al., 2020) | **95.37 ± 0.14** | **95.01 ± 0.71** | **95.16 ± 0.19** | **95.23 ± 0.07** | **95.21 ± 0.14** | **92.56 ± 0.42** | 76.94 ± 0.22 | **71.13 ± 0.48** |
| Negative-LS (Wei et al., 2021) | **94.92 ± 0.25** | 91.97 ± 0.46 | 90.29 ± 0.32 | 90.37 ± 0.12 | 90.13 ± 0.19 | 82.99 ± 0.36 | **77.06 ± 0.73** | 58.59 ± 0.98 |
| JoCoR (Wei et al., 2020) | 93.40 ± 0.24 | 91.44 ± 0.05 | 90.30 ± 0.20 | 90.21 ± 0.19 | 90.11 ± 0.21 | 83.37 ± 0.30 | 74.07 ± 0.33 | 59.97 ± 0.24 |
| CORES$^2$ (Cheng et al., 2021) | 93.43 ± 0.24 | 91.23 ± 0.11 | 89.66 ± 0.32 | 89.91 ± 0.45 | 89.79 ± 0.50 | 83.60 ± 0.53 | 75.56 ± 0.53 | 61.15 ± 0.73 |
| CORES* (Cheng et al., 2021) | 94.16 ± 0.11 | **95.25 ± 0.09** | 94.45 ± 0.14 | 94.88 ± 0.31 | 94.74 ± 0.03 | 91.66 ± 0.09 | 73.87 ± 0.16 | 55.72 ± 0.42 |
| VolMinNet (Li et al., 2021) | 92.14 ± 0.30 | 89.70 ± 0.21 | 88.30 ± 0.12 | 88.27 ± 0.09 | 88.19 ± 0.41 | 80.53 ± 0.20 | 70.61 ± 0.88 | 57.80 ± 0.31 |
| CAL (Zhu et al., 2021a) | 94.50 ± 0.31 | 91.97 ± 0.32 | 90.93 ± 0.31 | 90.75 ± 0.30 | 90.74 ± 0.24 | 85.36 ± 0.16 | 75.67 ± 0.25 | 61.73 ± 0.42 |
| PES (Semi) (Bai et al., 2021) | 94.76 ± 0.2 | 94.66 ± 0.18 | **95.06 ± 0.15** | **95.19 ± 0.23** | **95.22 ± 0.13** | **92.68 ± 0.22** | **77.92 ± 0.04** | **70.36 ± 0.33** |
| SOP (Liu et al., 2022) | N/A | **95.61 ± 0.13** | **95.28 ± 0.13** | **95.31 ± 0.10** | **95.39 ± 0.11** | **93.24 ± 0.21** | N/A | **67.81 ± 0.23** |

## E.2  PERFORMANCE COMPARISONS ON SYNTHETIC CIFAR DATASETS

Continuing the empirical observations in Section 5.1, in this section, we adopt ResNet-34 (He et al., 2016), the same training procedure and batch-size for all implemented methods. The synthetic CIFAR datasets generate synthetic noisy labels by using the same class-dependent noise transition matrices in CIFAR-10N and CIFAR-100N. The comparisons of test accuracies are summarized in Table 9. For

most methods, class-dependent synthetic noise is much easier to learn on CIFAR-10, especially when the noise level is high. The performance difference between human noise and the synthetic noise is less obvious for CIFAR-100.

Table 9: Comparison of test accuracies (%) on CIFAR-10 and CIFAR-100 (fine-label) with synthetic noisy labels using different methods. Top 3 performances are highlighted in **bold** (mean±standard deviation of 5 runs).

| Method | CIFAR-10 synthetic | | | | | | CIFAR-100 synthetic | |
|---|---|---|---|---|---|---|---|---|
| | Clean | Aggregate | Random 1 | Random 2 | Random 3 | Worst | Clean | Noisy |
| CE (Standard) | 92.92 ± 0.11 | 92.12 ± 0.17 | 91.03 ± 0.64 | 90.96 ± 0.35 | 90.98 ± 0.32 | 86.69 ± 0.16 | 76.70 ± 0.74 | 56.70 ± 1.26 |
| Forward $T$ (Patrini et al., 2017) | 93.02 ± 0.12 | 92.54 ± 0.20 | **91.70 ± 0.22** | 91.00 ± 0.21 | 91.31 ± 0.19 | 86.87 ± 0.44 | 76.18 ± 0.37 | 56.87 ± 1.58 |
| Backward $T$ (Patrini et al., 2017) | 93.10 ± 0.05 | 92.31 ± 0.28 | 91.34 ± 0.16 | 91.14 ± 0.41 | 91.36 ± 0.14 | 86.80 ± 0.42 | 76.79 ± 0.60 | 57.68 ± 1.90 |
| GCE (Zhang & Sabuncu, 2018) | 92.83 ± 0.16 | 92.44 ± 0.21 | 91.38 ± 0.17 | 91.17 ± 0.07 | **91.49 ± 0.19** | **87.12 ± 0.16** | 76.35 ± 0.48 | 57.17 ± 1.51 |
| Co-teaching (Han et al., 2018) | 93.35 ± 0.14 | 91.57 ± 0.32 | 90.99 ± 0.27 | 90.97 ± 0.24 | 91.31 ± 0.14 | 85.74 ± 0.36 | 73.46 ± 0.09 | 59.63 ± 0.27 |
| Co-teaching+ (Yu et al., 2019) | 92.41 ± 0.20 | 91.50 ± 0.13 | 90.62 ± 0.16 | 90.33 ± 0.46 | 90.59 ± 0.06 | 85.89 ± 0.25 | 70.99 ± 0.22 | 57.27 ± 0.23 |
| T-Revision (Xia et al., 2019) | 93.35 ± 0.23 | 90.01 ± 0.21 | 88.59 ± 0.63 | 88.56 ± 0.88 | 88.22 ± 0.58 | 83.57 ± 0.68 | 72.83 ± 0.21 | 50.91 ± 1.00 |
| Peer Loss (Liu & Guo, 2020) | 93.99 ± 0.13 | 92.65 ± 0.12 | 91.48 ± 0.20 | **91.50 ± 0.14** | 90.52 ± 0.24 | 86.67 ± 0.19 | 74.67 ± 0.36 | 56.74 ± 0.70 |
| ELR (Liu et al., 2020) | 93.45 ± 0.65 | 91.60 ± 0.19 | 90.65 ± 0.03 | 90.64 ± 0.24 | 90.90 ± 0.26 | 81.94 ± 0.27 | 72.78 ± 0.01 | 59.99 ± 0.88 |
| ELR+ (Liu et al., 2020) | **95.39 ± 0.05** | **94.98 ± 0.15** | **94.77 ± 0.18** | **94.60 ± 0.05** | **94.69 ± 0.14** | **87.38 ± 2.66** | **78.57 ± 0.12** | **68.46 ± 0.07** |
| Positive-LS (Lukasik et al., 2020) | 94.77 ± 0.17 | 92.13 ± 0.15 | 91.02 ± 0.53 | 91.15 ± 0.16 | 91.30 ± 0.14 | 86.71 ± 0.34 | 76.25 ± 0.35 | 56.51 ± 0.71 |
| F-Div (Wei & Liu, 2020) | 94.88 ± 0.12 | 92.36 ± 0.41 | 91.32 ± 0.29 | 91.12 ± 0.46 | 91.20 ± 0.10 | 86.67 ± 0.38 | 76.14 ± 0.36 | 58.41 ± 0.90 |
| Divide-Mix (Li et al., 2020) | **95.36 ± 0.09** | **95.45 ± 0.09** | **95.58 ± 0.07** | **95.51 ± 0.08** | **95.50 ± 0.10** | **93.55 ± 0.40** | **76.94 ± 0.22** | **71.78 ± 0.28** |
| Negative-LS (Wei et al., 2021) | **94.92 ± 0.25** | **92.74 ± 0.25** | 91.60 ± 0.30 | 91.45 ± 0.28 | **91.49 ± 0.23** | 86.99 ± 0.14 | **77.06 ± 0.73** | 59.85 ± 1.15 |
| JoCoR (Wei et al., 2020) | 93.40 ± 0.24 | 91.79 ± 0.21 | 91.08 ± 0.20 | 90.89 ± 0.24 | 91.12 ± 0.20 | 85.80 ± 0.33 | 74.07 ± 0.33 | 59.49 ± 0.29 |
| CORES$^2$ (Cheng et al., 2021) | 93.43 ± 0.24 | 92.72 ± 0.21 | 91.35 ± 0.30 | 91.43 ± 0.28 | 91.45 ± 0.18 | 85.27 ± 0.63 | 75.56 ± 0.53 | 60.43 ± 1.20 |
| VolMinNet (Li et al., 2021) | 92.14 ± 0.30 | 90.64 ± 0.09 | 89.54 ± 0.12 | 89.41 ± 0.13 | 89.35 ± 0.15 | 83.58 ± 0.24 | 70.61 ± 0.88 | 59.65 ± 0.70 |
| CAL (Zhu et al., 2021a) | 94.50 ± 0.31 | 92.22 ± 0.21 | 90.97 ± 0.35 | 90.79 ± 0.24 | 90.83 ± 0.31 | 85.80 ± 0.33 | 75.67 ± 0.00 | **62.20 ± 0.67** |

## E.3 EXPERIMENT DETAILS ON CIFAR-10N AND CIFAR-100N

The basic hyper-parameters settings for CIFAR-10N and CIFAR-100N are listed as follows: mini-batch size (128), optimizer (SGD), initial learning rate (0.1), momentum (0.9), weight decay (0.0005), number of epochs (100) and learning rate decay (0.1 at 50 epochs). Standard data augmentation is applied to each dataset.

**Special treatments** In the reproduced experiments, we use the default setting for Cores* (Cheng et al., 2021), ELR+ (Liu et al., 2020) and DivideMix since either advanced data augmentation strategies or two deep neural networks are adopted. And we fix a same pre-trained model for methods that require a CE warm-up or a pre-trained model.

## E.4 COMPUTING INFRASTRUCTURE

All our experiments run on a GPU cluster (500 GPUs of all kinds, mainly use 2080 Ti) for training and evaluation.

