# OpenReview forum: "Learning with Noisy Labels Revisited: A Study Using Real-World Human Annotations"
_ICLR.cc/2022/Conference — ICLR 2022 Poster_

### Official Review · Reviewer_EZud · 2021-10-27

**Correctness:** 3
**Technical Novelty And Significance:** 3
**Empirical Novelty And Significance:** 3
**Recommendation:** 6
**Confidence:** 5

**Main Review:**

This paper has several strengths although there is no technical novelty. In particular, the analysis with human-annotated label noise (using CIFAR-10N and CIFAR-100N) provides some insightful information. However, I have some critical concerns below:

First of all, I agree with the absence of controllable real noisy data, which can be used popularly for fair comparison or in-depth analysis (satisfying the three conditions: resolution, clean labels, and no interventions). For this purpose, CIFAR-10N and CIFAR-100N will be good reference data for research. However, I wonder if the data with such a small resolution (32x32) can properly reflect real noise. In real-world applications, the resolution of training images is mostly larger than 32x32, and the corrupted labels happen even with high-resolution images (e.g., medical images or data with fine-grained classes). I think that the restriction of the low resolution (actually, 32x32 is not moderate resolution) can generate label noise attributed to ambiguity in the scene (e.g., color information could be dominant than appearance information in labeling). Therefore, the collected noisy labels may be different from the noise label in a real real-world scenario.

Second, the observation of imbalanced annotation is quite interesting. Owing to this imbalanced annotation, I think CIFAR-10N and CIFAR-100N have class imbalance problems. Can authors report the imbalance ratio of the two proposed datasets? If the ratio is low, the results in performance evaluation are convincing. However, if the ratio is high, the low performance of robust approaches can be due to the more complicated scenario of 'class imbalance + label noise'. Similarly, in all other analyses, I am wondering if the class imbalance factor was properly handled.

Third, many readers will be interested in why the existing approaches show poor performance on human noise. The table summarizes many values but I feel hard to reason why ELR achieves the best performance consistently. In addition, why does the DivideMix shows the worst performance in the CIFAR-10N aggregate dataset? There is no discussion and analysis on these points. I know, that human noise is like a feature-dependent noise, thus it is much harder than the synthetic noise setup. But, why do all methods show inconsistent trends on that?

These are some minor comments:
- Clothing1M data has a 38% noise level (see the paper [1], the authors reported the clean accuracy of labeling was 61.54%)
[1] Learning from massive noisy labeled data for image classification, CVPR'15
- The explanation of Figure 7 is very confusing. e.g., what is the 'wrong' prediction on 'wrong 'labels? and there is no f_M in the paper (see the caption for Figure 7) Is it f_H?


**Summary Of The Paper:**

This paper focuses on learning from noisy labels, which is a very important research topic in the ML community. One of the main challenges in this area has been the absence of controllable real-world noisy data. The authors present the variation of CIFARs data with human annotations via MTurk.  By comparing the results on the original CIFARs and the newly proposed CIFAR-N, four observations are detailed quantitatively or qualitatively. In addition, they provide the performance comparison of more than 10 approaches on CIFAR-10 and -100N. The analysis of the memorization effect in human-annotated noisy data is also discussed.

**Summary Of The Review:**

As a paper that proposes insightful analysis and some datasets, there is room for further improvement. In particular, resolving the above concerns will add more clarity and novelty to the paper. I will decide my final rating after the rebuttal.

> [rebuttal] The score increases 5 $\rightarrow$ 6 since the response addressed my concerns properly.

---

> ### Author Response · Authors · 2021-11-18
> **Response to Reviewer EZud**
>
> $\textbf{Dear Reviewer EZud,}$
>
> We sincerely appreciate your time in reviewing our work! And we address your concerns as below.
>
> **1. No technical novelty**
>
> **Response:**
>
> We do have technical novelty, for example,
> - We introduce a visualization method based on the group-dependent noise transition vectors;
> - Our hypothesis testing method of instance-dependent label noise provides a quantitative tool for testing the simulated human labels.
>
> But our contributions also include:
>
> - **New insights and observations:**
>   - Imbalance human annotations;
>   - Human annotators have different preferences for similar classes (i.e., mouse>shrew, bus>automobile);
>   - The real-world noise label transition matrix;
>   - Multi-label issues in CIFAR-100 training images;
>   - Most classes in CIFAR-100N are class-dependent while the remaining classes are feature dependent;
>   - The study of memorizing effects further reveals the difference between real-world noise and synthetic noise, etc.
>
> - **Benchmarking effort:** A comprehensive leaderboard of >15 existing robust methods under CIFAR-N and synthetic noisy CIFAR datasets.
>
> **2. Low-resolution is not equivalent to real-world noise**
>
> **Response:**
>
> We agree that the noisy pattern in low-resolution images may differ from that in high-resolution cases. Our work aims to **supplement** existing large-scale as well as high-resolution image annotation tasks.
>
> And we want to highlight that **low-resolution images take up a crucial part of real-world images**, popular scenarios of low-resolution images include:
>
> - Due to the computation power, the training images may be down-scaled;
> - Segmented images from multi-label images for classification tasks frequently have a low resolution;
> - Pictures/Images taken in the low-light environment or taken by street CCTV;
> - Aerial or satellite based images for building extraction from remotely sensed images may have a low-resolution due to spatiotemporally accessibility and cost, etc.
>
> Besides, human annotators are more likely to give wrong annotations due to similar/vogue features in low-resolution images. As widely adopted in the learning-with-label-noise literature, low-resolution CIFAR images are more suitable/acceptable for researchers to conduct extensive experiments, yet, the lack of real-world human annotations motivates us to contribute CIFAR-N to the literature.
>
> **3. The imbalance ratio of the two proposed datasets**
>
> **Response:**
>
> We think the ratio is low compared to recent approaches in learning with class-imbalanced (long-tailed) data with label noise. In the literature of class-imbalanced learning, the imbalance ratio is usually denoted by $r=\dfrac{n_{max}}{n_{min}}$ with $r>5$ or $r>10$. Actually, we visualized the imbalance ratio for CIFAR-10N and CIFAR-100N in the appendix. It is quite obvious that there exist imbalance annotations, but the ratio is not that high. We also summarize the statistics of the imbalance ratio as below:
>
> |Aggregate | Random 1 | Random 2 | Random 3 | Worst | CIFAR-100 Fine|
> |:-------------:|:-------------:|:-------------:|:-------------:|:-------------:|:-------------:|
> |$r=$1.27 | $r=$1.25 | $r=$1.23 | $r=$1.26 | $r=$1.50 | $r=$4.35|

---

> > ### Author Response · Authors · 2021-11-18
> > **Additional Response to Reviewer EZud**
> >
> > **We proceed to address the remaining concerns as below.**
> >
> > **4. Performances of ELR [1] and Divide-Mix [2]**
> >
> > **Response:**
> >
> > For a fair comparison, we adopt the default hyper-parameter for each method. Several methods may need additional fine-tuning w.r.t. real-world noisy labels.  Divide-Mix[2] uses Gaussian mixture models to separate clean and corrupted instances, which may not be the best choice in low-noise settings. In experiments, we fixed $\text{lambda}_u=25$ for all noise settings as specified in the official implementation. We do observe that by setting $\text{lambda}_u=0$ (as mentioned in the appendix of [2]), its performance on the low noise setting can gain a performance increase. We have updated the reproduced results in our revised version (highlighted in **Blue**). And we re-state the performance of Divide-Mix on CIFAR-10N here:
> >
> > |Clean | Aggregate | Random 1 | Random 2 | Random 3|
> > |:-----:|:-----:|:-----:|:-----:|:-----:|
> > |95.37 ± 0.12 | 95.01 ± 0.71 | 95.16 ± 0.19 | 95.23 ± 0.07 | 95.21 ± 0.14|
> >
> > Among reproduced methods, ELR+ [1], Divide-Mix [2], and CORES* [3] tend to be more robust to CIFAR-N noisy labels. The reason is mainly due to the use of either two networks (and mix-up data augmentation) or a semi-supervised learning manner. These advanced tricks have a significant impact on the model performance by referring to the performance gap between ELR and ELR+, or CORES and CORES*.
> >
> > We will maintain a public leaderboard where authors are encouraged to update their performances within fair constraints (i.e., same architecture, etc)  and another leaderboard that has no restriction on any reasonable experimental details in the near future.
> >
> > **5. Inconsistent trends on the performance gap between real noise and synthetic noise**
> >
> > **Response:**
> >
> > Even if the noise rates are the same, we highlight that images/features of human-annotated wrong labels might be **harder** to learn than synthetic wrong labels.
> >
> > This is best explained by their **differences in memorizing effects** (CIFAR-10): for images that are provided with a wrongly annotated label by referring to the ground-truth label, deep neural nets easily memorize wrong predictions, while they have no/low confidence in the correct predictions. In Table 3, most methods that are of large performance gaps concentrate on loss adjustments methods that have shown great potential for synthetic noise labels. Thus, it makes sense that they fail to work well in real-world noisy settings by referring to the loss design or additional requirements for fine-tuning on human noise.
> >
> > ELR is specially designed to **alleviate the early memorization behavior** of noisy labels. Note that models memorize (fit on) human noisy labels more easily as shown in Figure 7, that is why we think ELR has a better performance on human noise compared to the synthetic ones.
> >
> > While for CIFAR-100N, we observe that the performance of most methods only shows slight improvements compared to Cross-Entropy loss. Even though CIFAR-100N is a small-scale data, we want to emphasize that CIFAR-100N indeed imposes a challenging task for existing learning-with-label-noise solutions. And the performance gap is harder to analyze. As mentioned in Observation 4 in CIFAR-100, we believe that the impacts of the **multi-label issue** (estimated >3% train images have more than 1 label) and the **imbalance annotations** (r=4.35) differs in various methods. We aim to leave this for future work.
> >
> > **Minor comments: Additional explanations for Figure 7**
> >
> > Wrong labels on wrong predictions mean that for images that are of wrongly annotated human labels, the memorized label (prediction with high confidence) by referring to the ground-truth label is wrong. In figure 7, we are interested in visualizing the memorized labels among clean labels (correct human annotations) and wrong (annotated label is not equal to the ground-truth label) labels. As mentioned in Definition 2, the label that is of >0.95 model prediction confidence is viewed as memorized. By referring to the ground-truth labels, we can further split the memorized labels into “correct” and “wrong” by checking whether this memorized label matches the ground-truth label or not, respectively.
> >
> > **Thanks for pointing out typos. We have corrected all mentioned ones.**
> >
> > References
> >
> > [1] Liu, S., Niles-Weed, J., Razavian, N. and Fernandez-Granda, C., 2020. Early-Learning Regularization Prevents Memorization of Noisy Labels. [NeurIPS 2020]
> >
> > [2] Li, J., Socher, R., & Hoi, S. C. DivideMix: Learning with Noisy Labels as Semi-supervised Learning. [ICLR 2019]
> >
> > [3] Cheng, H., Zhu, Z., Li, X., Gong, Y., Sun, X., & Liu, Y. Learning with Instance-Dependent Label Noise: A Sample Sieve Approach. [ICLR 2020]

---

> > > ### Comment · Reviewer_EZud · 2021-11-23
> > > **Thanks for the response.**
> > >
> > > Thanks for the detailed response. In my review, I pointed out three major concerns: (1) the low resolution, (2) imbalance problem in data, and (3) in-depth analysis of the inconsistent performance under the human annotation.
> > >
> > > I think the revised paper and response are enough to resolve most of my concerns.
> > >
> > > For the first concern, I agree that there are many important scenarios with low-resolution images, e.g., the cropped objects for image classification. Still, I think those images have a resolution larger than 32x32 in most cases, but I believe the proposed data will inspire more future research for understanding real-world noise.
> > >
> > > For the second concern, I appreciate the imbalance ratio for CIFAR10/100 datasets. Since the ratio is very low, I think the results for CIFAR10 are convincing (I know 4.35 is much lower than the imbalance ratio of ImageNet-LT (> 250)
> > >
> > > For the last concern, I appreciate the updated results and discussion for DivideMix. I agree that DivideMix is very sensitive to the hyperparameter choice. Moreover, the explanation for other approaches (e.g., ELR) is not perfect but gives good insight to me.
> > >
> > > To sum up, the response addressed most of my concerns properly, thus I increase my score to 6.

---

> > > > ### Author Response · Authors · 2021-11-25
> > > > **Thanks for the quick reply**
> > > >
> > > > $\textbf{Dear Reviewer EZud,}$
> > > >
> > > > We sincerely appreciate your quick response. And thanks again for your time and efforts in reviewing our work.
> > > >
> > > > We admit that the specific reasons why most approaches incur the positive performance gap require more thorough study - we very much hope this observation can kick off future studies to further understand the difference in existing algorithms when facing real human-patterned label noise.
> > > >
> > > > Sincerely
> > > >
> > > > ICLR 2022 Conference Paper499 Authors

---

### Official Review · Reviewer_q5d1 · 2021-10-29

**Correctness:** 3
**Technical Novelty And Significance:** 2
**Empirical Novelty And Significance:** 2
**Recommendation:** 6
**Confidence:** 5

**Main Review:**

Strengths: The paper highlights an important issue in the literature of label-noise-learning: synthetic noise is prevalently used to evaluate methods. This paper shows that real-world noise pattern is different from synthetic noise, and the datasets proposed could be useful as a smaller-scale benchmark for future researchers. The paper is also well-written in general.

Weaknesses:
1. Even though the proposed datasets offer an effective benchmark for researchers to conduct smaller-scale experiments, its significance is not enough. There already exists larger-scale real-world noisy datasets such as Food-101 and WebVision. Because the ultimate goal of label-noise-learning is to learn from weakly-labeled web data, these existing web datasets are sufficient for model evaluation. The paper argues that these datasets do not have clean training labels. However, it should not be a problem because they do have a clean test set for fair evaluation. Therefore, in my opinion, the proposed datasets do not address a research gap.
2. Besides the dataset, the paper does not offer other significant contributions. It can be expected that real-world noise has different distribution compared to synthetic noise. As a dataset paper without any technical novelty, I do not find this paper to provide much new insights that could guide future research directions in this field.

**Summary Of The Paper:**

This paper proposes two smaller-sized datasets with real-world label noise. The datasets contain CIFAR-10 and CIFAR-100 images with human-annotated labels. The paper analysis the noise pattern, and benchmarks existing methods on these datasets.

**Summary Of The Review:**

While I find this paper to have a clear strength, the contribution is not enough to justify acceptance to ICLR. The dataset proposed is good but not crucial, and the paper does not provide enough new insights. This paper might have a better chance at other venues specific for dataset papers.

---

> ### Author Response · Authors · 2021-11-18
> **Response to Reviewer q5d1**
>
> $\textbf{Dear Reviewer q5d1},$
>
> We sincerely appreciate your time in reviewing our work! And we address your concerns as below.
>
> **1. Significance is not enough**
>
> **Response:**
>
> We respectfully disagree with Reviewer q5d1’s statement on "Existing web datasets are sufficient for model evaluation even if there are no clean labels for verification” and “Our proposed datasets do not address a research gap”. Our reasons come as follows:
>
> - **Small-scale data is necessary:** Although the ultimate goal of learning with label noise is to learn from weakly labeled web data,
>    - **We have not even fully understood learning with label noise in small-scale (and low-resolution) images**.
>    - And we want to highlight that **low-resolution images take up a crucial part of real-world images**, for example,
>       - In the low-resolution image classification tasks;
>       - Down-scaled images due to constrained computation power;
>       - Segmented/local images where one image contains multiple labels that may have a low-resolution, etc.
>   - Besides, **the label noise on small-scale data differs from that on the large-scale data in several aspects**, for example,
>       - Low-resolution images are harder for human workers to annotate such that it is possible more and different noisy labels are induced, with possibly different noise patterns;
>       - Deep neural networks memorize features in small-scale data more easily than that appeared in large-scale ones, leading to a very suitable and controlled scenario to study the impact of neural networks memorizing noisy labels, etc.
>   - **Factors that contribute to the method performances on a large scale are not controllable:** Note that a large-scale dataset requires extensive fine-tuning and high computation power, many proposed methods may not be able to find the most suitable parameters that show the best performance. Before realizing the ultimate goal, we may need to address the challenges in the small-scale noisy datasets firstly.
>
> - **Development and understanding of noisy labels (such as human annotations) require clean labels:** Clean labels for verification is necessary. The existence of clean labels can not only provide motivations and new understandings for the noise pattern, but also be beneficial for the evaluation of label-noise detection methods, and some other communities, such as crowd-sourcing of data annotations in computer vision.
>
> **2. No significant contributions**
>
> **Response:**
>
> We respectfully disagree with Reviewer q5d1 that our work has no significant contributions except for the dataset. Actually, we also have contributions and technical novelty including:
>
> - **New insights and observations:**
>   - Imbalance human annotations;
>   - Human annotators have different preferences for similar classes (i.e., mouse>shrew, bus>automobile);
>   - The real-world noise label transition matrix;
>   - Multi-label issues in CIFAR-100 training images;
>   - Most classes in CIFAR-100N are class-dependent while the remaining classes are feature dependent;
>   - The study of memorizing effects further reveals the difference between real-world noise and synthetic noise, etc.
>
> - **Technical novelty:**
>   - We introduce a visualization method based on the group-dependent noise transition vectors;
>   - Our hypothesis testing method of instance-dependent label noise provides a quantitative tool for testing the simulated human labels.
>
> - **Benchmarking effort:** A comprehensive leaderboard of >15 existing robust methods under CIFAR-N and synthetic noisy CIFAR datasets.
>
> Our contributions may have the following Broader Impacts:
>
> - **Crowd-sourcing of data annotations in computer vision:** CIFAR-N may be further used for proposing simulations of human annotations in crowd-sourcing, where the expenses of obtaining human annotations are often tremendous.
>
> - **A template for hypothesizing label noise patterns:** Our hypothesis testing method of instance-dependent label noise may provide a quantitative tool for testing the simulated human labels.
>
> - **Benchmarking effort is important:** Although learning from noisy labels has witnessed thriving developments, we often observed conflicting comparisons due to the randomness in the synthetic noisy labels. While there exist several datasets with real human noise, we view our contribution as complementary to existing ones, due to the elaborated reasons above. We believe benchmarking existing and population solutions is an important technical contribution to the community.
>
> - **Understanding real-world label noise:** Our observations and CIFAR-N help with understanding real-world label noise. Besides, our observations of the multi-label issues in CIFAR-100 impose a new label noise pattern that is largely neglected.
>
> - **Motivations for real-world label noise solutions:** Our observations, especially the memorizing effects of real-world label noise may provide the literature with motivations for addressing real-world label noise.

---

> > ### Comment · Reviewer_q5d1 · 2021-11-23
> > **Thanks for the response**
> >
> > I appreciate the detailed response. My major concerns are well addressed by the response, hence I would raise my score to 6.

---

> > > ### Author Response · Authors · 2021-11-25
> > > **Thanks for the quick reply**
> > >
> > > $\textbf{Dear Reviewer q5d1,}$
> > >
> > > We sincerely appreciate your quick response! And thanks again for your time and efforts in reviewing our work.
> > >
> > > Sincerely
> > >
> > > ICLR 2022 Conference Paper499 Authors

---

### Official Review · Reviewer_PLPg · 2021-10-31

**Correctness:** 4
**Technical Novelty And Significance:** 3
**Empirical Novelty And Significance:** 4
**Recommendation:** 8
**Confidence:** 4

**Main Review:**

Strengths:
- The paper is clear and well-written.
- The annotation collection procedure was clearly described.
- A large number of baselines were implemented and compared in Section 5.1.

Weaknesses:
- How does the annotation collection procedure differ from the original CIFAR10 & CIFAR100? Are there any important differences or modifications to the process the authors made? These points should be addressed.
- The "qualitative" section on instance-dependent noise comparisons is confusing and does not help the overall flow of the paper. Figure 5 is more confusing than helpful, and there is no real "conclusion" from the subjective analysis until we get to the quantitative section with the hypothesis test.

-- UPDATE after rebuttal --
The authors have sufficiently addressed my concerns regarding the differences with respect to CIFAR-10H. Thus, I will raise my score from 6 to 8 with the understanding that the authors will update the paper to include the information they provided in their rebuttal reply.

**Summary Of The Paper:**

This paper collects additional human annotations for Cifar10 and Cifar100. These annotations are also compared to synthetic label noise alternatives.

**Summary Of The Review:**

The paper collects additional manual annotations for CIFAR10 and CIFAR100, which will be of great value to the research community, for ex. helping practitioners study the robustness of their algorithms to label noise.

---

> ### Author Response · Authors · 2021-11-18
> **Response to Reviewer PLPg**
>
> $\textbf{Dear Reviewer PLPg},$
>
> We sincerely appreciate your positive review! And we address your concerns as below.
>
> **1. Differences in annotation collection procedure between CIFAR and CIFAR-N**
>
> **Response:**
>
> Thanks for this great question! In our revised version, we provide detailed information on the label collection procedure of CIFAR, and the comparisons of label collection between CIFAR and CIFAR-N in Appendix C, highlighted in blue.
>
> We summarize the label collection procedure of CIFAR dataset as below:
>
> - **Images sources:** the images of CIFAR come from down-scaled images from 80 million color images obtained by various search engines. The corresponding searching terms are viewed as a noisy source label/class;
>
> - **Remove irrelevant images:** the designer provided each student (labeler) with a class. Students were asked to verify all the images which were found with that class as the search term. And reject images that are outside of the assigned class.
>
> - **Payment:** the payment for each student/labeler per hour is fixed.
>
> - **Noise rate:** with additional personal verification, the noise rate of each class is negligible, especially for CIFAR-10.
>
> We highlight the **differences in label collection procedure** between CIFAR and CIFAR-N as below:
>
> - **Annotation method:** CIFAR adopted search engines such that they obtained images by specifying the class name (conditioned on the label, from label to feature); while CIFAR-N makes use of CIFAR images and pays workers for annotating labels (from feature to label).
>
> - **Payment and incentives:** CIFAR fixed the payment for student workers (self-collected), there is no incentive to rush. As for CIFAR-N, we did not have any constraints on the workers via Amazon Mechanical Turk, i.e., the salary, the education level, etc. Besides, we set time limits and bonuses for each annotation task, there are incentives to rush and meanwhile provide high-quality labels.
>
> - **Noise rate:** for CIFAR, the noise rate can be viewed as 0. While for CIFAR-N, we provide several noisy label sets which are of various noise rates.
>
> **2. Additional explanations for Section 4.2.1**
>
> **Response:**
>
> Sorry for the confusion. Section 4.2.1 serves as a basis for understanding the hypothesis testing from the quantitative aspect in Section 4.2.2: for example, the definition of M-NN noise clusterability, transition vectors for each cluster among each class. We use the heatmap to visualize the transition vectors $p_{i, v}$ of cluster $v$ in class $i$. And the distance $d_{i, v}$ appeared in Section 4.2.2 comes from the square of $L_2$ distance
> - between transition vectors of human noise and synthetic noise ($d^{(1)}_{i, v}$);
> - between transition vectors of synthetic noise and synthetic noise ($d^{(2)}_{i, v}$).

---

> > ### Comment · Reviewer_PLPg · 2021-11-23
> > **Thanks for the response**
> >
> > Thanks for the detailed response. I appreciate the additional clarification between the original CIFAR and CIFAR-N data collection. I would also appreciate a more detailed comparison with Cifar-10H (in my followup reply) as well.

---

> ### Comment · Reviewer_PLPg · 2021-11-24
> **Comparison to Cifar-10H**
>
> (Had initially written this a week ago, but the viewer permissions were not set correctly. I am reposting now, and would appreciate a response from the authors if still possible.)
>
> After pondering over this paper a bit more, I am wondering why the authors did not more explicitly compare to the very related dataset Cifar-10H? https://github.com/jcpeterson/cifar-10h
>
> They seem to have done a very similar study, albeit only for the cifar10 test set. However, in their study, they have trained models on the 10,000-size test set and evaluated on the clean train set. From this point of view, it seems like the observations in Section 4 could have been simply done on cifar-10h, and it strongly suggests the learning analysis done in Section 5 could have been done using cifar-10h as well.
>
> It seems like the main contribution is thus to add 50k more human annotations to Cifar10. However, the authors have not explicitly verbalized why the jump from 10k -> 50k annotations is important and why large parts of the analyses they did could not have been done on Cifar-10h.
>
> I would be grateful if the authors could respond to this point as well.

---

> > ### Author Response · Authors · 2021-11-25
> > **More discussions on the differences between CIFAR-10H and 10N & Our technical contributions.**
> >
> > $\textbf{Dear Reviewer PLPg,}$
> >
> > This is a great question! We should definitely include more discussions about why our analysis builds on CIFAR-N rather than CIFAR-10H in our next version. In this response, we will first summarize the main collection procedure of CIFAR-10H, and then move to address your concerns.
> >
> > **CIFAR-10H label collection on CIFAR-10 test data:**
> >
> > - **No time limit:** participants were asked to select the label of each image from the surrounding text with no time limit.
> > - **Controlled workload:** Each participant is assigned 200 images (20 per class);
> > - **Attention check (intervention):** Participants with a low accuracy (<75%) on (easy-to-recognize) images were removed.
> > - **Other minor differences:** Label positions are shuffled among participants;47-63 annotations per image; payment: $0.0075 / 10 images.
> >
> > We want to highlight that the central query for CIFAR-10H is to understand the benefits of uncertainty in human annotations to improve the generalization power of the trained model. Therefore, a number of controls and interventions were applied when building CIFAR-10H to control the human noise rates to be not excessive to disturb the above benefit study (details below). We believe this aspect renders the dataset not super appropriate for the relevant studies reported in Section 4 and 5. Our detailed reasons come as follows:
> >
> > **1. Intervened real-world human noise pattern**
> >
> > - **Purpose of the dataset construction:** CIFAR-10H targets to identify the benefits from increasing the richness of label distributions (hard label → soft label) for image classification tasks. The soft labels are constructed by human uncertainty. **CIFAR-10H may not fully reveal the real-world human annotation noise due to the check & removal procedure as we described above in attention check.** While we aim to study the real-world label noise pattern: we only reject uninformative and spamming annotation patterns (e.g., labeling every task as class 1) and we do not restrict the number of annotations required from different workers with different working efficiency. We accept submissions even if a worker has a moderate accuracy (e.g., <60%) and meanwhile reward workers that contribute a large number of annotations.
> >
> > - **Noise rate:** we randomly select the i-th (e.g., 1,2,...,10th) annotated label for each test image in CIFAR-10-H, and there are approximately 5% wrong labels in the annotation. In CIFAR-10N, the random noise rate is around 18%. For CIFAR-100N, the noise rate increased to 40.20%. **CIFAR-10H has a much smaller noise rate due to the control intervention, and due to different objects of the collection.** We believe that a very low noise rate may deviate from real-world human noise.
> >
> > Therefore, we think our collection might have better captured the real-world noise patterns.
> >
> > **2. Training data V.S. Test data**
> >
> > - Training on CIFAR-10 test data may **lead to a model performance drop**. It is reported that, when trained using the much smaller test data, the generalization accuracy on training data is only about **83%** [R1]. Note the standard training and testing on CIFAR-10 has an accuracy of about **93%**. With added label noise, the substantial drop of the number of training data limits the possibility of fully evaluating the potentials and properties of the competing benchmark methods (e.g., learning and showing some of the established theoretical properties of a particular method might require a sufficient number of training data).
> >
> > - Looking forward, we think it might be beneficial to let the learning-with-noisy-label community have an option of training using 50k training data and testing on the 10k test data, **the same and standard way as other learning communities have developed and evaluated algorithms using CIFAR-10 data**. As we benchmarked in Table 2, most of the existing works are tuned (e.g., pre-trained models for representation extraction for CIFAR-10, etc) for the training with 50k training images. This can **help the community better align and calibrate the progress, as compared to other learning tasks** (e.g., supervised learning, semi-supervised learning, etc).
> >
> > **References**
> >
> > [R1] Peterson, J.C., Battleday, R.M., Griffiths, T.L. and Russakovsky, O., 2019. Human uncertainty makes classification more robust. In Proceedings of the IEEE/CVF International Conference on Computer Vision (pp. 9617-9626).

---

> > > ### Author Response · Authors · 2021-11-25
> > > **Additional response to "Comparison to Cifar-10H"**
> > >
> > > **Continuing the above discussion:**
> > >
> > > Besides, we also want to highlight that: **we also have other novel contributions beyond CIFAR-10N**, for example, new insights and observations, technical novelties, benchmarking efforts. For more details, please take a look at our response to **Reviewer q5d1** (Point **2. No significant contributions**)
> > >
> > > **New insights/observations from CIFAR-100N**
> > >
> > > What is more, it is worth noting that our analyses are not only on CIFAR-10. We also conducted studies on CIFAR-100, where we captured many different observations between CIFAR-10N and CIFAR-100N, to name a few:
> > >
> > > - The **dominant** class might be the wrong class, see Figure 2 ; (Section 4.1, observation 2)
> > >
> > > - **More imbalanced annotation**: the imbalance ratio in CIFAR-10N is <1.5; while the imbalance ratio is around 4.5 in CIFAR-100N (inspired by **Reviewer EZud**, thank you!).
> > >
> > > - The **multi-label issue** in CIFAR-100N, see Figure 4. (Section 4.1, observation 4)
> > >
> > > **Hypothesis testing**
> > >
> > > Our hypothesis test provides a template for analyzing the noise pattern, which we view as a technical contribution too. We can certainly apply our method to CIFAR-10H to understand the noise pattern. But as we pointed out earlier, the CIFAR-10H noise rate was controlled and intervened by the collection process to make it small, so it is unclear how this affects the hypothesis testing results.
> > >
> > > Sincerely
> > >
> > > ICLR 2022 Conference Paper499 Authors

---

> > > > ### Comment · Reviewer_PLPg · 2021-11-26
> > > > **Thanks for the response**
> > > >
> > > > Thank you for the detailed response.
> > > >
> > > > It is interesting to note that the noise rate for CIFAR-10H is relatively low compared to 10N (5% vs 18%). I also agree that studying feature-dependent noise might have been harder at lower noise regimes.
> > > >
> > > > Regarding data size, I also agree with the authors that many methods have been developed at the 50k sample regime, and thus having a dataset of the same size allows easier plug and play.
> > > >
> > > > The authors have sufficiently addressed my concerns regarding the differences with respect to CIFAR-10H. Thus, I will raise my score from 6 to 8 with the understanding that the authors will update the paper to include the information they provided in their rebuttal reply.

---

> > > > > ### Author Response · Authors · 2021-11-26
> > > > > **Thanks again for your valuable suggestions**
> > > > >
> > > > > **Dear Reviewer PLPg,**
> > > > >
> > > > > Thanks for the quick response! And we sincerely appreciate this valuable suggestion ''add comparisons among CIFAR-10, CIFAR-10H, and CIFAR-N''. In our next version, we will definitely include detailed comparisons and highlight the contributions of these three works.
> > > > >
> > > > > Best,
> > > > >
> > > > > ICLR 2022 Conference Paper499 Authors

---

### Official Review · Reviewer_7fRY · 2021-11-01

**Correctness:** 4
**Technical Novelty And Significance:** 3
**Empirical Novelty And Significance:** 3
**Recommendation:** 8
**Confidence:** 5

**Main Review:**

Overall, the paper provides great value to the research community in not just learning from noisy labels, but also data labeling [1]. The paper is easy to follow, and the observations are comprehensive.

**Strength**
- The paper systematically compares human noise and synthetic noise (including class-dependent label noise and instance-dependent label noise) thoroughly.
- The paper compares human and synthetic labels in terms of the label distribution and the corresponding model.

**Weakness**
- (minor) The paper can make some connections with the worker simulations in data labeling. They also adopt a similar class-dependent scheme for worker simulations. [1,2, 3]


**Question**
- In the bottom row of figure 7, are the predictions compared to the given label (wrong labels) or the ground truth labels?

**Other**

Some observations (3 and 5) coincide with the observations in prior work [1]. Prior work [1] considers class-dependent label noise but initializes the label noise with crowdsourced human noise. It also observes that using the human label noise is more challenging in data labeling.

[1] Y. Liao, A. Kar, and S. Fidler. Towards good practices for efficiently annotating large-scale image classification datasets, CVPR2021

[2] G. Hua, C. Long, M. Yang, and Y. Gao. Collaborative active learning of a kernel machine ensemble for recognition, CVPR2013

[3] C. Long and G. Hua. Multi-class multi-annotator active learning with robust gaussian process for visual recognition, CVPR2015


**Summary Of The Paper:**

The paper provides an extensive study on 1) the difference between human annotations and synthetic ones and 2) the impact on the classifier learning from human annotations and synthetic ones. To this end, the paper crowdsources the human annotations for CIFAR10 and CIFAR100 from Amazon Mechanical Turk. The observations show that human annotations are more challenging to learn, and human annotations result in a different memorization scheme for neural networks.

**Summary Of The Review:**

The paper provides great value in the research communities that focus on the impact of label noise. The paper is well written and easy to follow. The observations shown in the paper are supported by extensive analysis. There are only some parts that require more explanation, but it does not harm the overall readability.
To make the paper more complete, I suggest the author make more connections toward a broader community that focus on label noise.

---

> ### Author Response · Authors · 2021-11-18
> **Response to Reviewer 7fRY**
>
> $\textbf{Dear Reviewer 7fRY,}$
>
> We sincerely appreciate your positive feedback! And we address your concerns as below.
>
> **1. Suggestion 1: discuss our connections with the worker simulations in data labeling**
>
> **Response:**
>
> We would like to thank Reviewer 7fRY for this valuable suggestion. We included more discussions about the worker simulations in data labeling in our revised version, and we highlighted the changes in **Blue** in the **Introduction** section and the **Broader Impacts** (Appendix).
>
> **2. Additional explanations of Figure 7**
>
> **Response:**
>
> In the bottom row of Figure 7, the predictions are compared to the **ground truth labels**: in Figure 7, we are interested in visualizing the fraction of memorized labels. As mentioned in Definition 2, the label is viewed as memorized when its model confidence is $>0.95$. By referring to the ground-truth labels, we can further split the memorized labels into “correct” and “wrong” by checking whether this memorized label matches the ground-truth label or not, respectively.
>
> **3. Connection with [1]**
>
> **Response:**
>
> We agree with Reviewer 7fRY that human noise tends out to be more challenging as mentioned in [1], i.e., learning from simulated workers with uniform noise results in a lower test accuracy than the corresponding synthetic noise. We want to highlight that Observation 5 complements with observations in [1], for example, depending on the dataset as well as the proposed robust methods, learning with real-world human noise may not incur a worse performance than learning with synthetic noise (such as ELR on CIFAR-10, and datasets with a large number of labels, i.e., on CIFAR-100). Thanks to **Reviewer EZud**’s good catch, we also include additional explanations about the performance gap between synthetic and real-world label noise in our response to Reviewer EZud (**Inconsistent trends on the performance gap between real noise and synthetic noise**).
>
> **4. Suggestion 2: include more connections towards a broader community**
>
> **Response:**
>
> We would like to thank Reviewer 7fRY for this valuable suggestion. We discussed the **Broader Impacts** of our work in our revised version at the beginning of the Appendix (highlighted in blue). We restate here:
>
> - **Crowd-sourcing of data annotations in computer vision:** CIFAR-N may be further used for studying/proposing simulations of human annotations in crowd-sourcing, where the expenses of obtaining human annotations are often tremendous.
>
> - **A template for hypothesizing label noise patterns:** Our hypothesis testing method of instance-dependent label noise may provide a quantitative tool for testing the simulated human labels.
>
> - **Benchmarking effort:** Although learning from noisy labels has witnessed thriving developments, we often observed conflicting comparisons due to the randomness in the synthetic noisy labels. While there exist several datasets with real human noise, we view our contribution as complementary to existing ones, due to the elaborated reasons above. We hope our benchmarking efforts could be beneficial for the learning-with-label-noise community.
>
> - **Understanding real-world label noise:** Our observations and the provided human-annotated labels help with understanding real-world label noise. Besides, our observations of the **multi-label issues** in CIFAR-100 impose a new label noise pattern that is largely neglected.
>
> - **Motivations for real-world label noise solutions:** Our observations, especially the memorizing effects of real-world label noise may provide the learning-with-label-noise literature with motivations for addressing real-world label noise.
>
> References
>
> [1] Y. Liao, A. Kar, and S. Fidler. Towards good practices for efficiently annotating large-scale image classification datasets. [CVPR 2021]

---

### Author Response · Authors · 2021-11-18
**Rebuttal revision of our paper is uploaded!**

$\textbf{Dear Reviewers and Readers,}$

We want to thank the reviewers for their constructive suggestions! The differences between the revised version and the first version are highlighted in **Blue**. To summarize,

- We included brief discussions about the worker simulations in data labeling in the **Introduction** section and **Broader Impacts** (Appendix).

- We updated the results of Divide-Mix [1] on CIFAR-10N (aggregate and random noise, as well as the corresponding synthetic noisy labels) as suggested in the Appendix of [1].

- At the beginning of the Appendix, we discussed the potential **Broader Impacts** of our observations and contributions.

- We included the **comparisons of label collection procedure** between CIFAR and CIFAR-N in Section C (Appendix).

- We highlighted **a new observation** at the end of Section D (Appendix): human-annotated noisy labels of approximately 50 classes in CIFAR-100N are indeed class-dependent, while the remaining classes are feature-dependent.

Thanks!

ICLR 2022 Conference Paper499 Authors

References

[1] Li, J., Socher, R., & Hoi, S. C. DivideMix: Learning with Noisy Labels as Semi-supervised Learning. [ICLR 2019]

---

### Decision · Program_Chairs · 2022-01-20

**Decision:**

Accept (Poster)

**Comment:**

The authors propose two new benchmark datasets CIFAR-10-N and CIFAR-100-N, variants of CIFAR-10 and CIFAR-100 with real-world human annotation noise. The benchmark datasets are more realistic (e.g. instance-dependent noise) than some existing synthetic benchmarks for label noise. The authors also benchmark several popular baselines on the proposed benchmark

All the reviewers thought that this is an useful contribution to the community and appreciated the detailed author response. The consensus decision leaned towards accept. I recommend acceptance & encourage the authors to address any remaining concerns in the final version.
Please clarify the license (e.g. MIT license) when you release the dataset.